# HIQL: Offline Goal-Conditioned RL
# with Latent States as Actions

**Seohong Park**[1]    **Dibya Ghosh**[1]    **Benjamin Eysenbach**[2]    **Sergey Levine**[1]
[1]University of California, Berkeley    [2]Princeton University
`seohong@berkeley.edu`

## Abstract

Unsupervised pre-training has recently become the bedrock for computer vision and natural language processing. In reinforcement learning (RL), goal-conditioned RL can potentially provide an analogous self-supervised approach for making use of large quantities of unlabeled (reward-free) data. However, building effective algorithms for goal-conditioned RL that can learn directly from diverse offline data is challenging, because it is hard to accurately estimate the exact value function for faraway goals. Nonetheless, goal-reaching problems exhibit structure, such that reaching distant goals entails first passing through closer subgoals. This structure can be very useful, as assessing the quality of actions for nearby goals is typically easier than for more distant goals. Based on this idea, we propose a hierarchical algorithm for goal-conditioned RL from offline data. Using one action-free value function, we learn two policies that allow us to exploit this structure: a high-level policy that treats states as actions and predicts (a latent representation of) a subgoal and a low-level policy that predicts the action for reaching this subgoal. Through analysis and didactic examples, we show how this hierarchical decomposition makes our method robust to noise in the estimated value function. We then apply our method to offline goal-reaching benchmarks, showing that our method can solve long-horizon tasks that stymie prior methods, can scale to high-dimensional image observations, and can readily make use of action-free data. Our code is available at https://seohong.me/projects/hiql/

## 1 Introduction

Many of the most successful machine learning systems for computer vision [15, 36] and natural language processing [10, 18] leverage large amounts of unlabeled or weakly-labeled data. In the reinforcement learning (RL) setting, offline goal-conditioned RL provides an analogous way to potentially leverage large amounts of multi-task data without reward labels or video data without action labels: offline learning [54, 55] enables leveraging previously collected and passively observed data, and goal-conditioned RL [44, 79] enables learning from unlabeled, reward-free data. However, offline goal-conditioned RL poses major challenges. First, learning an accurate goal-conditioned value function for any state and goal pair is challenging when considering very broad and long-horizon goal-reaching tasks. This often results in a noisy value function and thus potentially an erroneous policy. Second, while the offline setting unlocks the potential for using previously collected data, it is not straightforward to incorporate vast quantities of existing action-free video data into standard RL methods. In this work, we aim to address these challenges by developing an effective offline goal-conditioned RL method that can learn to reach distant goals, readily make use of data without reward labels, and even utilize data without actions.

One straightforward approach to offline goal-conditioned RL is to first train a goal-conditioned value function and then train a policy that leads to states with high values. However, many prior papers have observed that goal-conditioned RL is very difficult, particularly when combined with offline training and distant goals [35, 38, 102]. We observe that part of this difficulty stems from the "signal-to-noise"

37th Conference on Neural Information Processing Systems (NeurIPS 2023).

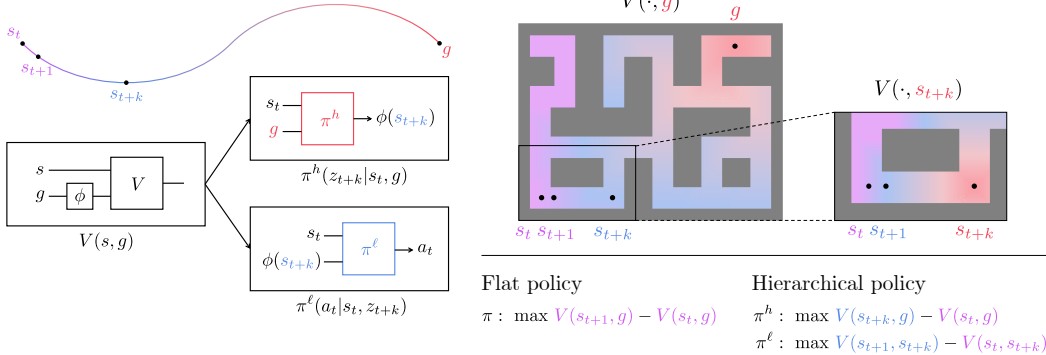

(a) **Three components of HIQL.**      (b) **Hierarchical policies get clearer learning signals.**

Figure 1: *(left)* We train a value function parameterized as $V(s, \phi(g))$, where $\phi(g)$ corresponds to the subgoal representation. The high-level policy predicts the representation of a subgoal $z_{t+k} = \phi(s_{t+k})$. The low-level policy takes this representation as input to produce actions to reach the subgoal. *(right)* In contrast to many prior works on hierarchical RL, we extract both policies from the *same* value function. Nonetheless, this hierarchical structure yields a better "signal-to-noise" ratio than a flat, non-hierarchical policy, due to the improved relative differences between values.

ratio in value functions for faraway goals: when the goal is far away, the optimal action may be only slightly better than suboptimal actions, because a transition in the wrong direction can simply be corrected at the next time step. Thus, when the value function is learned imperfectly and has small errors, these errors can drown out the signal for distant goals, potentially leading to an erroneous policy. This issue is further exacerbated with the offline RL setting, as erroneous predictions from the value function are not corrected when those actions are taken and their consequences observed.

To address this challenge, we separate policy extraction into two levels. We first train a goal-conditioned value function from offline data with implicit Q-learning (IQL) [49] and then we extract two-level policies from it. Our high-level policy produces intermediate waypoint states, or *subgoals*, as actions. Because predicting high-dimensional states can be challenging, we will propose a method that only requires the high-level policy to product *representations* of the subgoals, with the representations learned end-to-end from the value function. Our low-level policy takes this subgoal representation as input and produces actions to reach the subgoal (Figure 1a). Here, in contrast to previous hierarchical methods [56, 65], we extract both policies from the *same* value function. Nonetheless, this hierarchical decomposition enables the value function to provide clearer learning signals for both policies (Figure 1b). For the high-level policy, the value difference between various subgoals is much larger than that between different low-level actions. For the low-level policy, the value difference between actions becomes relatively larger because the low-level policy only needs to reach nearby subgoals. Moreover, the value function and high-level policy do not require action labels, so this hierarchical scheme provides a way to leverage a potentially large amount of passive, action-free data. Training the low-level policy does require some data labeled with actions.

To summarize, our main contribution in this paper is to propose **Hierarchical Implicit Q-Learning** (**HIQL**), a simple hierarchical method for offline goal-conditioned RL. HIQL extracts all the necessary components—a representation function, a high-level policy, and a low-level policy—from a single goal-conditioned value function. Through our experiments on six types of state-based and pixel-based offline goal-conditioned RL benchmarks, we demonstrate that HIQL significantly outperforms previous offline goal-conditioned RL methods, especially in complex, long-horizon tasks, scales to high-dimensional observations, and is capable of incorporating action-free data.

## 2 Related work

Our method draws on concepts from offline RL [54, 55], goal-conditioned RL [4, 44, 79], hierarchical RL [6, 62, 77, 85, 86, 96], and action-free RL [7, 12, 34, 80, 88, 105], providing a way to effectively train general-purpose goal-conditioned policies from previously collected offline data. Prior work on goal-conditioned RL has introduced algorithms based on a variety of techniques, such as hindsight relabeling [4, 13, 27, 56, 57, 75, 100], contrastive learning [23, 24, 102], and state-occupancy matching [20, 60].

However, directly solving goal-reaching tasks is often challenging in complex, long-horizon environments [35, 56, 65]. To address this issue, several goal-conditioned RL methods have been proposed based on hierarchical RL [11, 17, 51, 56, 65, 66, 81, 91, 103] or graph-based subgoal planning [22, 38, 40, 45, 46, 69, 78, 101]. Like these prior methods, our algorithm will use higher-level subgoals in a hierarchical policy structure, but we will focus on solving goal-reaching tasks from *offline* data. We use an offline RL algorithm [49] to train a goal-conditioned value function from the dataset, which allows us to simply *extract* the hierarchical policies in a decoupled manner with no need for potentially complex graph-based planning procedures. Another important difference from prior work is that we only train a *single* goal-conditioned value function, unlike previous hierarchical methods that train multiple hierarchical value functions [56, 65]. Perhaps surprisingly, we show that this can still significantly improve the performance of the hierarchical policies, due to an improved "signal-to-noise" ratio (Section 4).

Our method is most closely related to previous works on hierarchical offline skill extraction and hierarchical offline (goal-conditioned) RL. Offline skill extraction methods [2, 43, 50, 72, 76, 84] encode trajectory segments into a latent skill space, and learn to combine these skills to solve downstream tasks. The primary challenge in this setting is deciding how trajectories should be decomposed hierarchically, which can be sidestepped in our goal-conditioned setting since subgoals provide a natural decomposition. Among goal-conditioned approaches, hierarchical imitation learning [35, 59] jointly learns subgoals and low-level controllers from optimal demonstrations. These methods have two drawbacks: they predict subgoals in the raw observation space, and they require expert trajectories; our observation is that a value function can alleviate both challenges, as it provides a way to use suboptimal data and stitch across trajectories, as well as providing a latent goal representation in which subgoals may be predicted. Another class of methods plans through a graph or model to generate subgoals [25, 26, 58, 83]; our method simply extracts all levels of the hierarchy from a single unified value function, avoiding the high computational overhead of planning. Finally, our method is closely related to POR [97], which predicts the immediate next state as a subgoal; this can be seen as one extreme of our method without representations, although we show that more long-horizon subgoal prediction can be advantageous both in theory and practice.

## 3 Preliminaries

**Problem setting.** We consider the problem of offline goal-conditioned RL, defined by a Markov decision process $\mathcal{M} = (\mathcal{S}, \mathcal{A}, \mu, p, r)$ and a dataset $\mathcal{D}$, where $\mathcal{S}$ denotes the state space, $\mathcal{A}$ denotes the action space, $\mu \in \mathcal{P}(\mathcal{S})$ denotes an initial state distribution, $p \in \mathcal{S} \times \mathcal{A} \rightarrow \mathcal{P}(\mathcal{S})$ denotes a transition dynamics distribution, and $r(s, g)$ denotes a goal-conditioned reward function. The dataset $\mathcal{D}$ consists of trajectories $\tau = (s_0, a_0, s_1, a_1, \ldots, s_T)$. In some experiments, we assume that we have an additional action-free dataset $\mathcal{D}_\mathcal{S}$ that consists of state-only trajectories $\tau_s = (s_0, s_1, \ldots, s_T)$. Unlike some prior work [4, 40, 46, 65, 101], we assume that the goal space $\mathcal{G}$ is the same as the state space (*i.e.*, $\mathcal{G} = \mathcal{S}$). Our goal is to learn from $\mathcal{D} \cup \mathcal{D}_\mathcal{S}$ an optimal goal-conditioned policy $\pi(a|s, g)$ that maximizes $J(\pi) = \mathbb{E}_{g \sim p(g), \tau \sim p^\pi(\tau)}[\sum_{t=0}^{T} \gamma^t r(s_t, g)]$ with $p^\pi(\tau) = \mu(s_0) \prod_{t=0}^{T-1} \pi(a_t \mid s_t, g) p(s_{t+1} \mid s_t, a_t)$, where $\gamma$ is a discount factor and $p(g)$ is a goal distribution.

**Implicit Q-learning (IQL).** One of the main challenges with offline RL is that a policy can exploit overestimated values for out-of-distribution actions [55], as we cannot correct erroneous policies and values via environment interactions, unlike in online RL. To tackle this issue, Kostrikov et al. [49] proposed implicit Q-learning (IQL), which avoids querying out-of-sample actions by converting the $\max$ operator in the Bellman optimal equation into expectile regression. Specifically, IQL trains an action-value function $Q_{\theta_Q}(s, a)$ and a state-value function $V_{\theta_V}(s)$ with the following loss:

$$\mathcal{L}_V(\theta_V) = \mathbb{E}_{(s,a) \sim \mathcal{D}}[L_2^\tau(Q_{\bar{\theta}_Q}(s, a) - V_{\theta_V}(s))], \tag{1}$$

$$\mathcal{L}_Q(\theta_Q) = \mathbb{E}_{(s,a,s') \sim \mathcal{D}}[(r_{\text{task}}(s, a) + \gamma V_{\theta_V}(s') - Q_{\theta_Q}(s, a))^2], \tag{2}$$

where $r_{\text{task}}(s, a)$ denotes the task reward function, $\bar{\theta}_Q$ denotes the parameters of the target Q network [64], and $L_2^\tau$ is the expectile loss with a parameter $\tau \in [0.5, 1)$: $L_2^\tau(x) = |\tau - \mathbb{1}(x < 0)|x^2$. Intuitively, expectile regression can be interpreted as an asymmetric square loss that penalizes positive values more than negative ones. As a result, when $\tau$ tends to 1, $V_{\theta_V}(s)$ gets closer to $\max_a Q_{\bar{\theta}_Q}(s, a)$ (Equation (1)). Thus, we can use the value function to estimate the TD target $(r_{\text{task}}(s, a) + \gamma \max_{a'} Q_{\bar{\theta}_Q}(s', a'))$ as $(r_{\text{task}}(s, a) + \gamma V_{\theta_V}(s'))$ without having to sample actions $a'$.

After training the value function with Equations (1) and (2), IQL extracts the policy with advantage-weighted regression (AWR) [67, 70, 71, 73, 74, 94]:

$$J_\pi(\theta_\pi) = \mathbb{E}_{(s,a,s')\sim\mathcal{D}}[\exp(\beta \cdot (Q_{\bar{\theta}_Q}(s,a) - V_{\theta_V}(s))) \log \pi_{\theta_\pi}(a \mid s)], \qquad (3)$$

where $\beta \in \mathbb{R}_0^+$ denotes an inverse temperature parameter. Intuitively, Equation (3) encourages the policy to select actions that lead to large $Q$ values while not deviating far from the data collection policy [71].

**Action-free goal-conditioned IQL.** The original IQL method described above requires both reward and action labels in the offline data to train the value functions via Equations (1) and (2). However, in real-world scenarios, offline data might not contain task information or action labels, as in the case of task-agnostic demonstrations or videos. As such, we focus on the setting of offline goal-conditioned RL, which does not require task rewards, and provides us with a way to incorporate state-only trajectories into value learning. We can use the following action-free variant [34, 97] of IQL to learn an offline goal-conditioned value function $V_{\theta_V}(s, g)$:

$$\mathcal{L}_V(\theta_V) = \mathbb{E}_{(s,s')\sim\mathcal{D}_S, g\sim p(g|\tau)}[L_2^\tau(r(s,g) + \gamma V_{\bar{\theta}_V}(s',g) - V_{\theta_V}(s,g))]. \qquad (4)$$

Unlike Equations (1) and (2), this objective does not require actions when fitting the value function, as it directly takes backups from the values of the next states.

Action-labeled data is only needed when extracting the policy. With the goal-conditioned value function learned by Equation (4), we can extract the policy with the following variant of AWR:

$$J_\pi(\theta_\pi) = \mathbb{E}_{(s,a,s')\sim\mathcal{D}, g\sim p(g|\tau)}[\exp(\beta \cdot A(s,a,g)) \log \pi_{\theta_\pi}(a \mid s, g)], \qquad (5)$$

where we approximate $A(s, a, g)$ as $\gamma V_{\theta_V}(s', g) + r(s, g) - V_{\theta_V}(s, g)$. Intuitively, Equation (5) encourages the policy to select the actions that lead to the states having high values. With this action-free variant of IQL, we can train an optimal goal-conditioned value function only using action-free data and extract the policy from action-labeled data that may be different from the passive dataset.

We note that this action-free variant of IQL is unbiased when the environment dynamics are deterministic [34], but it may overestimate values in stochastic environments. This deterministic environment assumption is inevitable for learning an unbiased value function solely from state trajectories. The reason is subtle but important: in stochastic environments, it is impossible to tell whether a good outcome was caused by taking a good action or because of noise in the environment. As a result, applying action-free IQL to stochastic environments will typically result in overestimating the value function, implicitly assuming that all noise is controllable. While we will build our method upon Equation (4) in this work for simplicity, in line with many prior works on offline RL that employ similar assumptions [14, 33, 34, 41, 42, 93, 97], we believe correctly handling stochastic environments with advanced techniques (*e.g.*, by identifying controllable parts of the environment [92, 99]) is an interesting direction for future work.

# 4   Hierarchical policy structure for offline goal-conditioned RL

Goal-conditioned offline RL provides a general framework for learning flexible policies from data, but the goal-conditioned setting also presents an especially difficult multi-task learning problem for RL algorithms, particularly for long-horizon tasks where the goal is far away. In Section 4.1, we discuss some possible reasons for this difficulty, from the perspective of the "signal-to-noise" ratio in the learned goal-conditioned value function. We then propose hierarchical policy extraction as a solution (Section 4.2) and compare the performances of hierarchical and flat policies in a didactic environment, based on our theoretical analysis (Section 4.3).

## 4.1   Motivation: why non-hierarchical policies might struggle

One common strategy in offline RL is to first fit a value function and then extract a policy that takes actions leading to high values [3, 8, 29–31, 49, 52, 67, 71, 95, 97, 98, 100]. This strategy can be directly applied to offline goal-conditioned RL by learning a goal-conditioned policy $\pi(a \mid s_t, g)$ that aims to maximize the learned goal-conditioned value function $V(s_{t+1}, g)$, as in Equation (5). However, when the goal $g$ is far from the current state $s$, the learned goal-conditioned value function may not provide a clear learning signal for a flat, non-hierarchical policy. There are two reasons for this. First, the differences between the values of different next states ($V(s_{t+1}, g)$) may be small, as bad outcomes by taking suboptimal actions may be simply corrected in the next few steps, causing

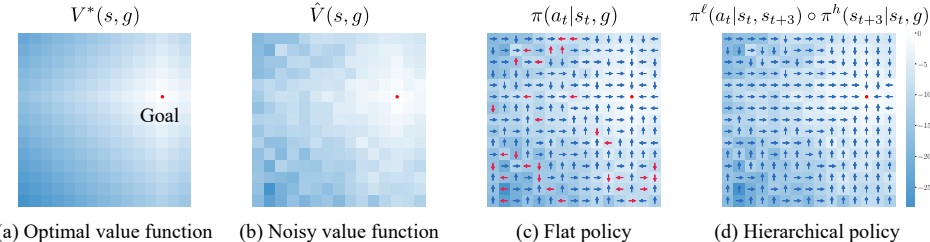

| $V^*(s,g)$ | $\hat{V}(s,g)$ | $\pi(a_t|s_t,g)$ | $\pi^\ell(a_t|s_t,s_{t+3}) \circ \pi^h(s_{t+3}|s_t,g)$ |
|:---:|:---:|:---:|:---:|
| (a) Optimal value function | (b) Noisy value function | (c) Flat policy | (d) Hierarchical policy |

Figure 2: **Hierarchies allow us to better make use of noisy value estimates.** *(a)* In this gridworld environment, the optimal value function predicts higher values for states $s$ that are closer to the goal $g$ (•). *(b, c)* However, a noisy value function results in selecting incorrect actions ($\rightarrow$). *(d)* Our method uses this *same* noisy value function to first predict an intermediate subgoal, and then select an action for reaching this subgoal. Actions selected in this way correctly lead to the goal.

only relatively minor costs. Second, these small differences can be overshadowed by the noise present in the learned value function (due to, for example, sampling error or approximation error), especially when the goal is distant from the current state, in which case the magnitude of the goal-conditioned value (and thus the magnitude of its noise or errors) is large. In other words, the "signal-to-noise" ratio in the next time step values $V(s_{t+1},g)$ can be small, not providing sufficiently clear learning signals for the flat policy. Figure 2 illustrates this problem. Figure 2a shows the ground-truth optimal value function $V^*(s,g)$ for a given goal at each state, which can guide the agent to reach the goal. However, when noise is present in the learned value function $\hat{V}(s,g)$ (Figure 2b), the flat policy $\pi(a \mid s,g)$ becomes erroneous, especially at states far from the goal (Figure 2c).

## 4.2   Our hierarchical policy structure

To address this issue, our main idea in this work, which we present fully in Section 5, is to separate policy extraction into two levels. Instead of directly learning a single, flat, goal-conditioned policy $\pi(a \mid s_t, g)$ that aims to maximize $V(s_{t+1}, g)$, we extract both a high-level policy $\pi^h(s_{t+k} \mid s_t, g)$ and a low-level policy $\pi^\ell(a \mid s_t, s_{t+k})$, which aims to maximize $V(s_{t+k}, g)$ and $V(s_{t+1}, s_{t+k})$, respectively. Here, $s_{t+k}$ can be viewed as a waypoint or *subgoal*. The high-level policy outputs intermediate subgoal states that are $k$ steps away from $s$, while the low-level policy produces primitive actions to reach these subgoals. Although we extract both policies from the *same* learned value function in this way, this hierarchical scheme provides clearer learning signals for both policies. Intuitively, the high-level policy receives a more reliable learning signal because different subgoals lead to more dissimilar values than primitive actions. The low-level policy also gets a clear signal (from the same value function) since it queries the value function with only nearby states, for which the value function is relatively more accurate (Figure 1b). As a result, the overall hierarchical policy can be more robust to noise and errors in the value function (Figure 2d).

## 4.3   Didactic example: hierarchical policies mitigate the signal-to-noise ratio challenge

To further understand the benefits of hierarchical policies, we study a toy example with one-dimensional state space (Figure 3). In this environment, the agent can move one unit to the left or right at each time step. The agent gets a reward of 0 when it reaches the goal; otherwise, it always gets $-1$. The optimal goal-conditioned value function is hence given as $V^*(s,g) = -|s-g|$ (assuming $\gamma = 1$). We assume that the noise in the learned value function $\hat{V}(s,g)$ is proportional to the optimal value: *i.e.*, $\hat{V}(s,g) = V^*(s,g) + \sigma z_{s,g} V^*(s,g)$, where $z_{s,g}$ is sampled independently from the standard normal distribution and $\sigma$ is its standard deviation. This indicates that as the goal becomes more distant, the noise generally increases, a trend we observed in our experiments (see Figure 8).

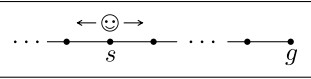

Figure 3: **1-D toy environment.**

In this scenario, we compare the probabilities of choosing incorrect actions under the flat and hierarchical policies. We assume that the distance between $s$ and $g$ is $T$ (*i.e.*, $g = s + T$ and $T > 1$). Both the flat policy and the low-level policy of the hierarchical approach consider the goal-conditioned values at $s \pm 1$. The high-level policy evaluates the values at $s \pm k$, using $k$-step away subgoals. For the hierarchical approach, we query both the high- and low-level policies at every step. Given these settings, we can bound the error probabilities of both approaches as follows:

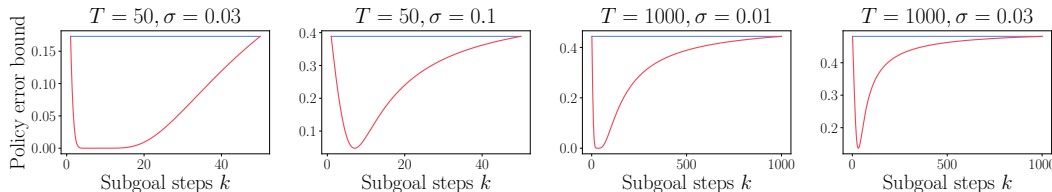

Figure 4: **Comparison of policy errors in flat vs. hierarchical policies in didactic environments.** The hierarchical policy, with an appropriate subgoal step, often yields significantly lower errors than the flat policy.

**Proposition 4.1.** *In the environment described in Figure 3, the probability of the flat policy $\pi$ selecting an incorrect action is given as $\mathcal{E}(\pi) = \Phi\left(-\frac{\sqrt{2}}{\sigma\sqrt{T^2+1}}\right)$ and the probability of the hierarchical policy $\pi^\ell \circ \pi^h$ selecting an incorrect action is bounded as $\mathcal{E}(\pi^\ell \circ \pi^h) \leq \Phi\left(-\frac{\sqrt{2}}{\sigma\sqrt{(T/k)^2+1}}\right) + \Phi\left(-\frac{\sqrt{2}}{\sigma\sqrt{k^2+1}}\right)$, where $\Phi$ denotes the cumulative distribution function of the standard normal distribution, $\Phi(x) = \mathbb{P}[z \leq x] = \frac{1}{\sqrt{2\pi}}\int_{-\infty}^{x} e^{-t^2/2}\mathrm{d}t.$*

The proof can be found in Appendix E.1. We first note that each of the error terms in the hierarchical policy bound is always no larger than the error in the flat policy, implying that both the high- and low-level policies are more accurate than the flat policy. To compare the total errors, $\mathcal{E}(\pi)$ and $\mathcal{E}(\pi^\ell \circ \pi^h)$, we perform a numerical analysis. Figure 4 shows the hierarchical policy's error bound for varying subgoal steps in five different $(T, \sigma)$ settings. The results indicate that the flat policy's error can be significantly reduced by employing a hierarchical policy with an appropriate choice of $k$, suggesting that splitting policy extraction into two levels can be beneficial.

# 5 Hierarchical Implicit Q-Learning (HIQL)

Based on the hierarchical policy structure in Section 4, we now present a practical algorithm, which we call **Hierarchical Implicit Q-Learning** (**HIQL**), to extract hierarchical policies that are robust to the noise present in the learned goal-conditioned value function. We first explain how to train a subgoal policy (Section 5.1) and then extend this policy to predict representations (learned via the value function), which will enable HIQL to scale to image-based environments (Section 5.2).

## 5.1 Hierarchical policy extraction

As motivated in Section 4.2, we split policy learning into two levels, with a high-level policy generating intermediate subgoals and a low-level policy producing primitive actions to reach the subgoals. In this way, the learned goal-conditioned value function can provide clearer signals for both policies, effectively reducing the total policy error. Our method, HIQL, extracts the hierarchical policies from the *same* value function learned by action-free IQL (Equation (4)) using AWR-style objectives. While we choose to use action-free IQL in this work, we note that our hierarchical policy extraction scheme is orthogonal to the choice of the underlying offline RL algorithm used to train a goal-conditioned value function.

HIQL trains both a high-level policy $\pi^h_{\theta_h}(s_{t+k} \mid s_t, g)$, which produces optimal $k$-step subgoals $s_{t+k}$, and a low-level policy $\pi^\ell_{\theta_\ell}(a \mid s_t, s_{t+k})$, which outputs primitive actions, with the following objectives:

$$J_{\pi^h}(\theta_h) = \mathbb{E}_{(s_t, s_{t+k}, g)}[\exp(\beta \cdot \tilde{A}^h(s_t, s_{t+k}, g)) \log \pi^h_{\theta_h}(s_{t+k} \mid s_t, g)], \tag{6}$$

$$J_{\pi^\ell}(\theta_\ell) = \mathbb{E}_{(s_t, a_t, s_{t+1}, s_{t+k})}[\exp(\beta \cdot \tilde{A}^\ell(s_t, a_t, s_{t+k})) \log \pi^\ell_{\theta_\ell}(a_t \mid s_t, s_{t+k})], \tag{7}$$

where $\beta$ denotes the inverse temperature hyperparameter and we approximate $\tilde{A}^h(s_t, s_{t+k}, g)$ as $V_{\theta_V}(s_{t+k}, g) - V_{\theta_V}(s_t, g)$ and $\tilde{A}^\ell(s_t, a_t, s_{t+k})$ as $V_{\theta_V}(s_{t+1}, s_{t+k}) - V_{\theta_V}(s_t, s_{t+k})$. We do not include rewards and discount factors in these advantage estimates for simplicity, as they are (mostly) constants or can be subsumed into the temperature $\beta$ (see Appendix A for further discussion). Similarly to vanilla AWR (Equation (5)), our high-level objective (Equation (6)) performs a weighted regression over subgoals to reach the goal, and the low-level objective (Equation (7)) carries out a weighted regression over primitive actions to reach the subgoals.

---

**Algorithm 1** Hierarchical Implicit Q-Learning (HIQL)

---

1: **Input**: offline dataset $\mathcal{D}$, action-free dataset $\mathcal{D}_\mathcal{S}$ (optional, $\mathcal{D}_\mathcal{S} = \mathcal{D}$ otherwise)
2: Initialize value function $V_{\theta_V}(s, \phi(g))$ with built-in representation $\phi(g)$, high-level policy $\pi^h_{\theta_h}(z_{t+k} \mid s_t, g)$,
   low-level policy $\pi^\ell_{\theta_\ell}(a \mid s_t, z_{t+k})$, learning rates $\lambda_V, \lambda_h, \lambda_\ell$
3: **while** not converged **do**
4:     $\theta_V \leftarrow \theta_V - \lambda_V \nabla_{\theta_V} \mathcal{L}_V(\theta_V)$ with $(s_t, s_{t+1}, g) \sim \mathcal{D}_\mathcal{S}$   # Train value function, Equation (4)
5: **end while**
6: **while** not converged **do**
7:     $\theta_h \leftarrow \theta_h + \lambda_h \nabla_{\theta_h} J_{\pi^h}(\theta_h)$ with $(s_t, s_{t+k}, g) \sim \mathcal{D}_\mathcal{S}$   # Extract high-level policy, Equation (6)
8: **end while**
9: **while** not converged **do**
10:     $\theta_\ell \leftarrow \theta_\ell + \lambda_\ell \nabla_{\theta_\ell} J_{\pi^\ell}(\theta_\ell)$ with $(s_t, a_t, s_{t+1}, s_{t+k}) \sim \mathcal{D}$   # Extract low-level policy, Equation (7)
11: **end while**

---

We note that Equation (6) and Equation (7) are completely separated from one another, and only the low-level objective requires action labels. As a result, we can leverage action-free data for both the value function and high-level policy of HIQL, by further training them with a potentially large amount of additional passive data. Moreover, the low-level policy is relatively easy to learn compared to the other components, as it only needs to reach local subgoals without the need for learning the complete global structure. This enables HIQL to work well even with a limited amount of action information, as we will demonstrate in Section 6.4.

## 5.2 Representations for subgoals

In high-dimensional domains, such as pixel-based environments, directly predicting subgoals can be prohibitive or infeasible for the high-level policy. To resolve this issue, we incorporate representation learning into HIQL, letting the high-level policy produce more compact *representations* of subgoals. While one can employ existing action-free representation learning methods [34, 61, 68, 82] to learn state representations, HIQL simply uses an intermediate layer of the value function as a goal representation, which can be proven to be sufficient for control. Specifically, we parameterize the goal-conditioned value function $V(s, g)$ with $V(s, \phi(g))$, and use $\phi(g)$ as the representation of the goal. Using this representation, the high-level policy $\pi^h(z_{t+k} \mid s_t, g)$ produces $z_{t+k} = \phi(s_{t+k})$ instead of $s_{t+k}$, which the low-level policy $\pi^\ell(a \mid s_t, z_{t+k})$ takes as input to output actions (Figure 1a). In this way, we can simply learn compact goal representations that are sufficient for control with no separate training objectives or components. Formally, we prove that the representations from the value function are sufficient for action selection:

**Proposition 5.1** (Goal representations from the value function are sufficient for action selection)**.** *Let $V^*(s, g)$ be the value function for the optimal reward-maximizing policy $\pi^*(a \mid s, g)$ in a deterministic MDP. Let a representation function $\phi(g)$ be given. If this same value function can be represented in terms of goal representations $\phi(g)$, then the reward-maximizing policy can also be represented in terms of goal representations $\phi(g)$:*

$$\exists \, V_\phi(s, \phi(g)) \; s.t. \; V_\phi(s, \phi(g)) = V^*(s, g) \, for \, all \, s, g \implies$$
$$\exists \, \pi_\phi(a \mid s, \phi(g)) \; s.t. \; \pi_\phi(a \mid s, \phi(g)) = \pi^*(a \mid s, g) \, for \, all \, s, g.$$

While Proposition 5.1 shows that the parameterized value function $V(s, \phi(g))$ provides a sufficient goal representation $\phi$, we found that additionally concatenating $s$ to the input to $\phi$ (*i.e.*, using $\phi([g, s])$ instead of $\phi(g)$) [39] leads to better empirical performance (see Appendix A for details), and thus we use the concatenated variant of value function parameterization in our experiments. We provide a pseudocode for HIQL in Algorithm 1 and the full training details in Appendices A and D.

## 6 Experiments

Our experiments will use six offline goal-conditioned tasks, aiming to answer the following questions:

1. How well does HIQL perform on a variety of goal-conditioned tasks, compared to prior methods?
2. Can HIQL solve image-based tasks, and are goal representations important for good performance?
3. Can HIQL utilize action-free data to accelerate learning?
4. Does HIQL mitigate policy errors caused by noisy and imperfect value functions in practice?

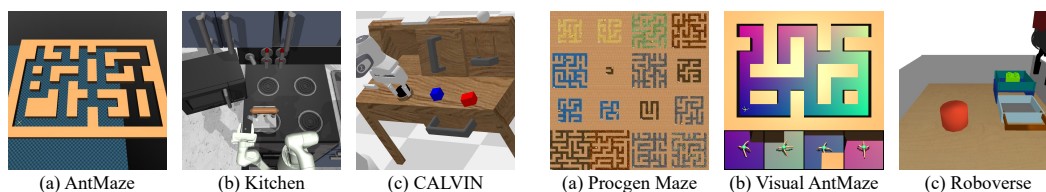

| (a) AntMaze | (b) Kitchen | (c) CALVIN |   | (a) Procgen Maze | (b) Visual AntMaze | (c) Roboverse |

Figure 5: **State-based benchmark environments.**    Figure 6: **Pixel-based benchmark environments.**

## 6.1 Experimental setup

We first describe our evaluation environments, shown in Figure 5 (state-based) and Figure 6 (pixel-based). **AntMaze** [9, 87] is a class of challenging long-horizon navigation tasks, where the goal is to control an 8-DoF Ant robot to reach a given goal location from the initial position. We use the four medium and large maze datasets from the original D4RL benchmark [28]. While the large mazes already present a significant challenge for long-horizon reasoning, we also include two even larger mazes (AntMaze-Ultra) proposed by Jiang et al. [43]. **Kitchen** [35] is a long-horizon manipulation domain, in which the goal is to complete four subtasks (*e.g.*, open the microwave or move the kettle) with a 9-DoF Franka robot. We employ two datasets consisting of diverse behaviors ('-partial' and '-mixed') from the D4RL benchmark [28]. **CALVIN** [63], another long-horizon manipulation environment, also features four target subtasks similar to Kitchen. However, the dataset accompanying CALVIN [84] consists of a much larger number of task-agnostic trajectories from 34 different subtasks, which makes it challenging for the agent to learn relevant behaviors for the goal. **Procgen Maze** [16] is a pixel-based maze navigation environment. We train agents on an offline dataset consisting of 500 or 1000 different maze levels with a variety of sizes, colors, and difficulties, and test them on both the same and different sets of levels to evaluate their generalization capabilities. **Visual AntMaze** is a vision-based variant of the AntMaze-Large environment [28]. We provide only a $64 \times 64 \times 3$ camera image (as shown in the bottom row of Figure 6b) and the agent's proprioceptive states, excluding the global coordinates. As such, the agent must learn to navigate the maze based on the wall structure and floor color from the image. **Roboverse** [25, 104] is a pixel-based, goal-conditioned robotic manipulation environment. The dataset consists of $48 \times 48 \times 3$ images of diverse sequential manipulation behaviors, starting from randomized initial object poses. We evaluate the agent's performance across five unseen goal-reaching tasks that require multi-stage reasoning and generalization. To train goal-conditioned policies in these benchmark environments, during training, we replace the original rewards with a sparse goal-conditioned reward function, $r(s, g) = 0$ (if $s = g$), $-1$ (otherwise).

We compare the performance of HIQL with six previous behavioral cloning and offline RL methods. For behavioral cloning methods, we consider flat goal-conditioned behavioral cloning (GCBC) [19, 33] and hierarchical goal-conditioned behavioral cloning (HGCBC) with two-level policies [35, 59]. For offline goal-conditioned RL methods, we evaluate a goal-conditioned variant of IQL [49] ("GC-IQL") (Section 3), which does not use hierarchy, and POR [97] ("GC-POR"), which uses hierarchy but does not use temporal abstraction (*i.e.*, similar to $k = 1$ in HIQL) nor representation learning. In AntMaze, we additionally compare HIQL with two model-based approaches that studied this domain in prior work: Trajectory Transformer (TT) [41], which models entire trajectories with a Transformer [90], and TAP [43], which encodes trajectory segments with VQ-VAE [89] and performs model-based planning over latent vectors in a hierarchical manner. We use the performance reported by Jiang et al. [43] for comparisons with TT and TAP. In our experiments, we use 8 random seeds and represent 95% confidence intervals with shaded regions (in figures) or standard deviations (in tables), unless otherwise stated. We provide full details of environments and baselines in Appendix D.

## 6.2 Results on state-based environments

We first evaluate HIQL in the five state-based environments (AntMaze-{Medium, Large, Ultra}, Kitchen, and CALVIN) using nine offline datasets. We evaluate the performance of the learned policies by commanding them with the evaluation goal state $g$ (*i.e.*, the benchmark task target position in AntMaze, or the state that corresponds to completing all four subtasks in Kitchen and CALVIN), and measuring the average return with respect to the original benchmark task reward function. We test two versions of HIQL (without and with representations) in state-based environments. Table 1 and Figure 7a show the results on the nine offline datasets, indicating that HIQL mostly achieves the best performance in our experiments. Notably, HIQL attains an 88% success rate on AntMaze-Large and 53% on AntMaze-Ultra, which is, to the best of our knowledge, better than any previously reported

Table 1: **Evaluating HIQL on state-based offline goal-conditioned RL.** HIQL mostly outperforms six baselines on a variety of benchmark tasks, including on different types of data. We show the standard deviations across 8 random seeds and refer to Appendix B for the full training curves. Baselines: GCBC [33], HGCBC [35], GC-IQL [49], GC-POR [97], TAP [43], TT [41].

| Dataset | GCBC | HGCBC | GC-IQL | GC-POR | TAP | TT | HIQL (ours) | HIQL (w/o repr.) |
|---|---|---|---|---|---|---|---|---|
| antmaze-medium-diverse | $67.3_{\pm10.1}$ | $71.6_{\pm8.9}$ | $63.5_{\pm14.6}$ | $74.8_{\pm11.9}$ | $85.0$ | $\mathbf{100.0}$ | $86.8_{\pm4.6}$ | $89.9_{\pm3.5}$ |
| antmaze-medium-play | $71.9_{\pm16.2}$ | $66.3_{\pm9.2}$ | $70.9_{\pm11.2}$ | $71.4_{\pm10.9}$ | $78.0$ | $\mathbf{93.3}$ | $84.1_{\pm10.8}$ | $87.0_{\pm8.4}$ |
| antmaze-large-diverse | $20.2_{\pm9.1}$ | $63.9_{\pm10.4}$ | $50.7_{\pm18.8}$ | $49.0_{\pm17.2}$ | $82.0$ | $60.0$ | $\mathbf{88.2}_{\pm5.3}$ | $87.3_{\pm3.7}$ |
| antmaze-large-play | $23.1_{\pm15.6}$ | $64.7_{\pm14.5}$ | $56.5_{\pm14.4}$ | $63.2_{\pm16.1}$ | $74.0$ | $66.7$ | $\mathbf{86.1}_{\pm7.5}$ | $81.2_{\pm6.6}$ |
| antmaze-ultra-diverse | $14.4_{\pm9.7}$ | $39.4_{\pm20.6}$ | $21.6_{\pm15.2}$ | $29.8_{\pm13.6}$ | $26.0$ | $33.3$ | $\mathbf{52.9}_{\pm17.4}$ | $52.6_{\pm8.7}$ |
| antmaze-ultra-play | $20.7_{\pm9.7}$ | $38.2_{\pm18.1}$ | $29.8_{\pm12.4}$ | $31.0_{\pm19.4}$ | $22.0$ | $20.0$ | $39.2_{\pm14.8}$ | $\mathbf{56.0}_{\pm12.4}$ |
| kitchen-partial | $38.5_{\pm11.8}$ | $32.0_{\pm16.7}$ | $39.2_{\pm13.5}$ | $18.4_{\pm14.3}$ | - | - | $\mathbf{65.0}_{\pm9.2}$ | $46.3_{\pm8.6}$ |
| kitchen-mixed | $46.7_{\pm20.1}$ | $46.8_{\pm17.6}$ | $51.3_{\pm12.8}$ | $27.9_{\pm17.9}$ | - | - | $\mathbf{67.7}_{\pm6.8}$ | $36.8_{\pm20.1}$ |
| calvin | $17.3_{\pm14.8}$ | $3.1_{\pm8.8}$ | $7.8_{\pm17.6}$ | $12.4_{\pm18.6}$ | - | - | $\mathbf{43.8}_{\pm39.5}$ | $23.4_{\pm27.1}$ |

Table 2: **Evaluating HIQL on pixel-based offline goal-conditioned RL.** HIQL scales to high-dimensional pixel-based environments with latent subgoal representations, achieving the best performance across the environments. We refer to Appendix B for the full training curves.

| Dataset | GCBC | HGCBC (+ repr.) | GC-IQL | GC-POR (+ repr.) | HIQL (ours) |
|---|---|---|---|---|---|
| procgen-maze-500-train | $16.8_{\pm2.8}$ | $14.3_{\pm4.1}$ | $72.5_{\pm10.0}$ | $75.8_{\pm12.1}$ | $\mathbf{82.5}_{\pm6.0}$ |
| procgen-maze-500-test | $14.5_{\pm5.0}$ | $11.2_{\pm3.7}$ | $49.5_{\pm9.8}$ | $53.8_{\pm14.5}$ | $\mathbf{64.5}_{\pm13.2}$ |
| procgen-maze-1000-train | $27.2_{\pm8.9}$ | $15.0_{\pm5.7}$ | $78.2_{\pm7.2}$ | $82.0_{\pm6.5}$ | $\mathbf{87.0}_{\pm13.9}$ |
| procgen-maze-1000-test | $12.0_{\pm5.9}$ | $14.5_{\pm5.0}$ | $60.0_{\pm10.6}$ | $69.8_{\pm7.4}$ | $\mathbf{78.2}_{\pm17.9}$ |
| visual-antmaze-diverse | $71.4_{\pm6.0}$ | $35.1_{\pm12.0}$ | $72.6_{\pm5.9}$ | $47.4_{\pm17.6}$ | $\mathbf{80.5}_{\pm9.4}$ |
| visual-antmaze-play | $64.4_{\pm6.3}$ | $23.8_{\pm8.5}$ | $70.4_{\pm26.6}$ | $57.0_{\pm8.1}$ | $\mathbf{78.4}_{\pm4.6}$ |
| visual-antmaze-navigate | $33.2_{\pm7.9}$ | $21.4_{\pm4.6}$ | $22.1_{\pm14.1}$ | $16.1_{\pm15.2}$ | $\mathbf{45.7}_{\pm18.1}$ |
| roboverse | $26.2_{\pm4.5}$ | $26.4_{\pm6.4}$ | $31.2_{\pm8.7}$ | $46.6_{\pm7.4}$ | $\mathbf{61.5}_{\pm5.3}$ |

(a) State-based environments    (b) Pixel-based environments

Figure 7: **Performance comparison.** Following the protocol proposed by Agarwal et al. [1], we report the interquartile mean (IQM) metrics to evaluate the statistical significance of HIQL's performance on offline goal-conditioned RL benchmarks (Tables 1 and 2). We refer to Appendix B for the full Rliable plots.

result on these datasets. In manipulation domains, we find that having latent subgoal representations in HIQL is important for enabling good performance. In CALVIN, while other methods often fail to achieve any of the subtasks due to the high diversity in the data, HIQL completes approximately two subtasks on average.

## 6.3 Results on pixel-based environments

Next, to verify whether HIQL can scale to high-dimensional environments using goal representations, we evaluate our method on three pixel-based domains (Procgen Maze, Visual AntMaze, and Robo-verse) with image observations. For the prior hierarchical approaches that generate raw subgoals (HGCBC and GC-POR), we apply HIQL's value-based representation learning scheme to enable them to handle the high-dimensional observation space. Table 2 and Figure 7b present the results, showing that our hierarchical policy extraction scheme, combined with representation learning, improves performance in these image-based environments as well. Notably, in Procgen Maze, HIQL exhibits larger gaps compared to the previous methods on the test sets. This is likely because the high-level policy can generalize better than the flat policy, as it can focus on the long-term direction toward the goal rather than the maze's detailed layout. In Roboverse, HIQL is capable of generalizing to solve unseen robotic manipulation tasks purely from images, achieving an average success rate of 62%.

## 6.4 Results with action-free data

As mentioned in Section 5.1, one of the advantages of HIQL is its ability to leverage a potentially large amount of passive (action-free) data. To empirically verify this capability, we train HIQL on action-limited datasets, where we provide action labels for just 25% of the trajectories and use state-only trajectories for the remaining 75%. Table 3 shows the results from six different tasks, demonstrating

Table 3: **HIQL can leverage passive, action-free data.** Since our method requires action information only for the low-level policy, which is relatively easier to learn, HIQL mostly achieves comparable performance with just 25% of action-labeled data, outperforming even baselines trained on full datasets.

| Dataset | GC-IQL (full) | GC-POR (full) | HIQL (full) | HIQL (action-limited) | vs. HIQL (full) | vs. Prev. best (full) |
|---|---|---|---|---|---|---|
| antmaze-large-diverse | 50.7 ±18.8 | 49.0 ±17.2 | 88.2 ±5.3 | 88.9 ±6.4 | +0.7 | +38.2 |
| antmaze-ultra-diverse | 21.6 ±15.2 | 29.8 ±13.6 | 52.9 ±17.4 | 38.2 ±15.4 | −14.7 | +8.4 |
| kitchen-mixed | 51.3 ±12.8 | 27.9 ±17.9 | 67.7 ±6.8 | 59.1 ±9.6 | −8.6 | +7.8 |
| calvin | 7.8 ±17.6 | 12.4 ±18.6 | 43.8 ±39.5 | 35.8 ±30.7 | −8.0 | +23.4 |
| procgen-maze-500-train | 72.5 ±10.0 | 75.8 ±12.1 | 82.5 ±6.0 | 77.0 ±12.5 | −5.5 | +1.2 |
| procgen-maze-500-test | 49.5 ±9.8 | 53.8 ±14.5 | 64.5 ±13.2 | 65.5 ±16.4 | +1.0 | +11.7 |

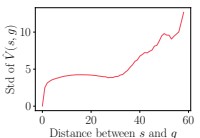
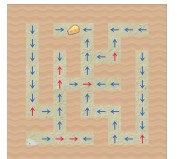

| Policy accuracy metric | GC-IQL | GC-POR (+ repr.) | HIQL (ours) |
|---|---|---|---|
| All goals (train) | 61.9 ±1.9 | 64.5 ±2.7 | **66.6** ±2.1 (+2.1) |
| All goals (test) | 60.3 ±2.8 | 63.6 ±3.2 | **68.0** ±4.1 (+4.4) |
| Distant goals (train) | 49.3 ±3.3 | 48.1 ±4.5 | **56.8** ±8.9 (+7.5) |
| Distant goals (test) | 47.5 ±8.6 | 47.2 ±3.7 | **59.9** ±10.4 (+12.4) |

Figure 8: **Value and policy errors in Procgen Maze**: *(left)* As the distance between the state and the goal increases, the learned value function becomes noisier. *(middle)* We measure the accuracies of learned policies. *(right)* Thanks to our hierarchical policy extraction scheme (Section 4.2), HIQL exhibits the best policy accuracy, especially when the goal is far away from the state. The blue numbers denote the accuracy differences between HIQL and the second-best methods.

that HIQL, even with a limited amount of action information, can mostly maintain its original performance. Notably, action-limited HIQL still outperforms previous offline RL methods (GC-IQL and GC-POR) trained with the full action-labeled data. We believe this is because HIQL learns a majority of the knowledge through hierarchical subgoal prediction from state-only trajectories.

### 6.5 Does HIQL mitigate policy errors caused by noisy value functions?

To empirically verify whether our two-level policy architecture is more robust to errors in the learned value function (*i.e.*, the "signal-to-noise" ratio argument in Section 4), we compare the policy accuracies of GC-IQL (flat policy), GC-POR (hierarchy without temporal abstraction), and HIQL (ours) in Procgen Maze, by evaluating the ratio at which the ground-truth actions match the learned actions. We also measure the noisiness (*i.e.*, standard deviation) of the learned value function with respect to the ground-truth distance between the state and the goal. Figure 8 shows the results. We first observe that the noise in the value function generally becomes larger as the state-goal distance increases. Consequently, HIQL achieves the best policy accuracy, especially for distant goals ($\text{dist}(s, g) \geq 50$), as its hierarchical policy extraction scheme provides the policies with clearer learning signals (Section 4.2).

We refer to Appendix C for further analyses, including **subgoal visualizations** and an **ablation study** on subgoal steps and design choices for representations.

## 7 Conclusion

We proposed HIQL as a simple yet effective hierarchical algorithm for offline goal-conditioned RL. While hierarchical RL methods tend to be complex, involving many different components and objectives, HIQL shows that it is possible to build a method where a single value function simultaneously drives the learning of the low-level policy, the high-level policy, and the representations in a relatively simple and easy-to-train framework. We showed that HIQL not only exhibits strong performance in various challenging goal-conditioned tasks, but also can leverage action-free data and enjoy the benefits of built-in representation learning for image-based tasks.

**Limitations.** One limitation of HIQL is that the objective for its action-free value function (Equation (4)) is unbiased only when the environment dynamics are deterministic. As discussed in Section 3, HIQL (and other prior methods that use action-free videos) may overestimate the value function in partially observed or stochastic settings. To mitigate the optimism bias of HIQL in stochastic environments, we believe disentangling controllable parts from uncontrollable parts of the environment can be one possible solution [92, 99], which we leave for future work. Another limitation of our work is that we assume the noise for each $V(s, g)$ is independent in our theoretical analysis (Proposition 4.1). While Figure 8 shows that the "signal-to-noise" argument empirically holds in our experiments, the independence assumption in our theorem might not hold in environments with continuous state spaces, especially when the value function is modeled by a smooth function approximator.

## Acknowledgments

We would like to thank Aviral Kumar for an informative discussion about the initial theoretical results, Chongyi Zheng, Kuan Fang, and Fangchen Liu for helping set up the Roboverse environment, and RAIL members and anonymous reviewers for their helpful comments. This work was supported by the Korea Foundation for Advanced Studies (KFAS), the Fannie and John Hertz Foundation, the NSF GRFP (DGE2140739), AFOSR (FA9550-22-1-0273), and ONR (N00014-21-1-2838). This research used the Savio computational cluster resource provided by the Berkeley Research Computing program at UC Berkeley.

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

## A    Training details

**Goal distributions.**    We train our goal-conditioned value function, high-level policy, and low-level policy respectively with Equations (4), (6) and (7), using different goal-sampling distributions. For the value function (Equation (4)), we sample the goals from either random states, futures states, or the current state with probabilities of 0.3, 0.5, and 0.2, respectively, following Ghosh et al. [34]. We use $\mathrm{Geom}(1-\gamma)$ for the future state distribution and the uniform distribution over the offline dataset for sampling random states. For the hierarchical policies, we mostly follow the sampling strategy of Gupta et al. [35]. We first sample a trajectory $(s_0, s_1, \ldots, s_t, \ldots, s_T)$ from the dataset $\mathcal{D_S}$ and a state $s_t$ from the trajectory. For the high-level policy (Equation (6)), we either (*i*) sample $g$ uniformly from the future states $s_{t_g}$ ($t_g > t$) in the trajectory and set the target subgoal to $s_{\min(t+k,t_g)}$ or (*ii*) sample $g$ uniformly from the dataset and set the target subgoal to $s_{\min(t+k,T)}$. For the low-level policy (Equation (7)), we first sample a state $s_t$ from $\mathcal{D}$, and set the input subgoal to $s_{\min(t+k,T)}$ in the same trajectory.

**Advantage estimates.**    In principle, the advantage estimates for Equations (6) and (7) are respectively given as

$$A^h(s_t, s_{t+\tilde{k}}, g) = \gamma^{\tilde{k}} V_{\theta_V}(s_{t+\tilde{k}}, g) + \sum_{t'=t}^{\tilde{k}-1} r(s_{t'}, g) - V_{\theta_V}(s_t, g), \tag{8}$$

$$A^\ell(s_t, a_t, \tilde{s}_{t+k}) = \gamma V_{\theta_V}(s_{t+1}, \tilde{s}_{t+k}) + r(s_t, \tilde{s}_{t+k}) - V_{\theta_V}(s_t, \tilde{s}_{t+k}), \tag{9}$$

where we use the notations $\tilde{k}$ and $\tilde{s}_{t+k}$ to incorporate the edge cases discussed in the previous paragraph (*i.e.*, $\tilde{k} = \min(k, t_g - t)$ when we sample $g$ from future states, $\tilde{k} = \min(k, T - t)$ when we sample $g$ from random states, and $\tilde{s}_{t+k} = s_{\min(t+k,T)}$). Here, we note that $s_{t'} \neq g$ and $s_t \neq \tilde{s}_{t+k}$ always hold except for those edge cases. Thus, the reward terms in Equations (8) and (9) are mostly constants (under our reward function $r(s, g) = 0$ (if $s = g$), $-1$ (otherwise)), as are the third terms (with respect to the policy inputs). As such, we practically ignore these terms for simplicity, and this simplification further enables us to subsume the discount factors in the first terms into the temperature hyperparameter $\beta$. We hence use the following simplified advantage estimates, which we empirically found to lead to almost identical performances in our experiments:

$$\tilde{A}^h(s_t, s_{t+\tilde{k}}, g) = V_{\theta_V}(s_{t+\tilde{k}}, g) - V_{\theta_V}(s_t, g), \tag{10}$$

$$\tilde{A}^\ell(s_t, a_t, \tilde{s}_{t+k}) = V_{\theta_V}(s_{t+1}, \tilde{s}_{t+k}) - V_{\theta_V}(s_t, \tilde{s}_{t+k}). \tag{11}$$

**State representations.**    We model the output of the representation function $\phi(g)$ in $V(s, \phi(g))$ with a 10-dimensional latent vector and normalize the outputs of $\phi(g)$ [53]. Empirically, we found that concatenating $s$ to the input (*i.e.*, using $\phi([g, s])$ instead of $\phi(g)$, Figure 9), similarly to Hong et al. [39], improves performance in our experiments. While this might lose the sufficiency property of the representations (*i.e.*, Proposition 5.1), we found that the representations obtained in this way generally lead to better performance in practice, indicating that they still mostly preserve the goal information for control. We believe this is due to the

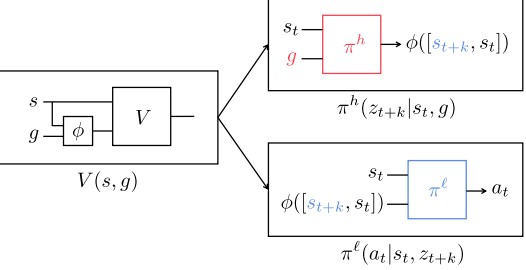

Figure 9: **Full architecture of HIQL.** In practice, we use $V(s, \phi([g, s]))$ instead of $V(s, \phi(g))$ as we found that the former leads to better empirical performance.

imposed bottleneck on $\phi$ by constraining its effective dimensionality to 9 (by using normalized 10-dimensional vectors), which enforces $\phi$ to retain bits regarding $g$ and to reference $s$ only when necessary. Additionally, in pixel-based environments, we found that allowing gradient flows from the low-level policy loss (Equation (7)) to $\phi$ further improves performance. We ablate these choices and report the results in Appendix C.

**Policy execution.**    At test time, we query both the high-level and low-level policies at every step, without temporal abstraction. We found that fixing subgoal states for more than one step does not significantly affect performance, so we do not use temporal abstraction for simplicity.

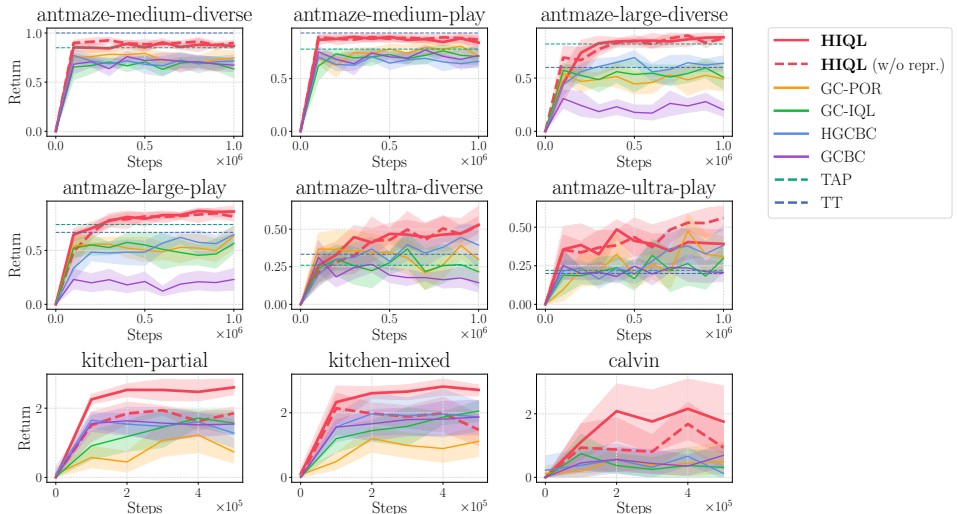

Figure 10: Training curves for the results with state-based environments (Table 1). Shaded regions denote the 95% confidence intervals across 8 random seeds.

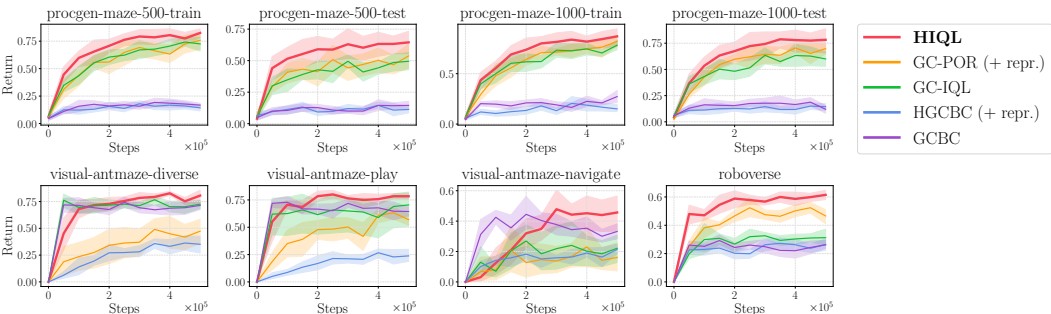

Figure 11: Training curves for the results with pixel-based environments (Table 2). Shaded regions denote the 95% confidence intervals across 8 random seeds.

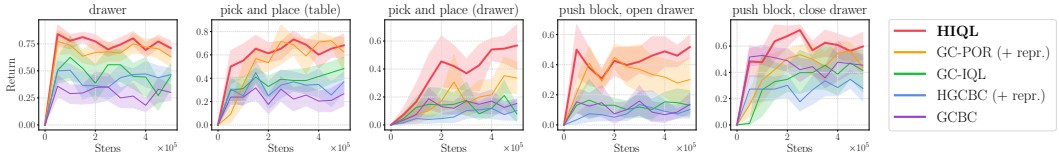

Figure 12: Training curves for the five tasks [104] in Roboverse. Shaded regions denote the 95% confidence intervals across 8 random seeds.

We provide a pseudocode for HIQL in Algorithm 1. We note that the high- and low-level policies can be jointly trained with the value function as well, as in Kostrikov et al. [49].

## B  Additional Plots

We include the training curves for Tables 1 to 3 in Figures 10, 11 and 13, respectively. We also provide the training curves for each of the five tasks [104] in Roboverse in Figure 12. We include the Rliable [1] plots in Figures 14 and 15. We note that the numbers in Tables 1 to 3 are *normalized* scores (see Appendix D), while the returns in the figures are unnormalized ones.

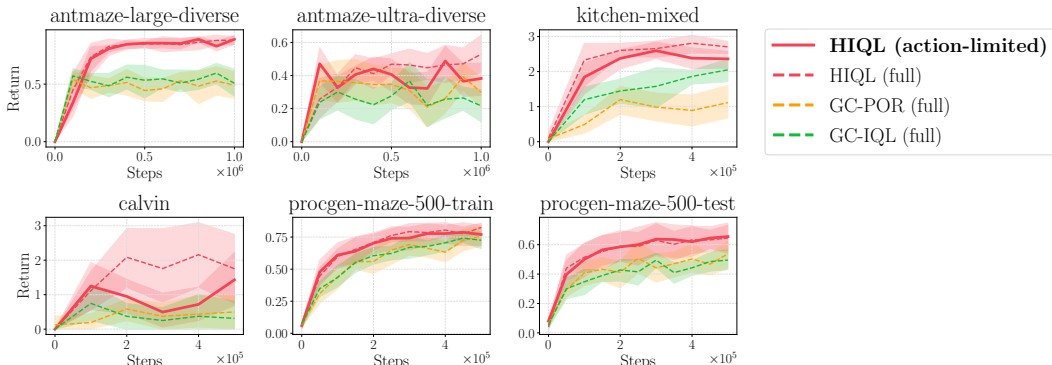

Figure 13: Training curves for the results with action-free data (Table 3). Shaded regions denote the 95% confidence intervals across 8 random seeds.

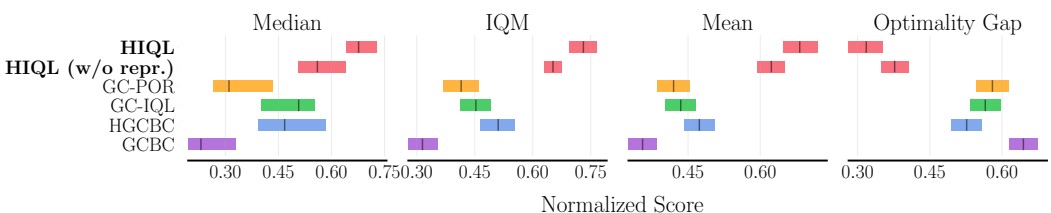

Figure 14: Rliable plots for state-based environments.

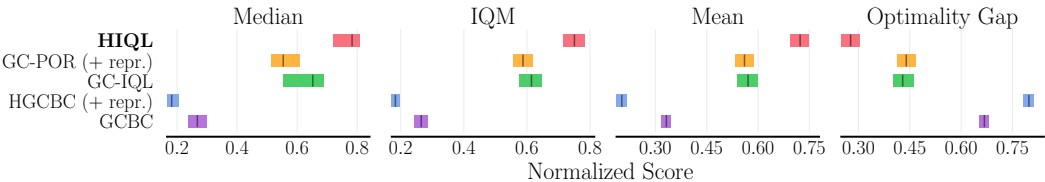

Figure 15: Rliable plots for pixel-based environments.

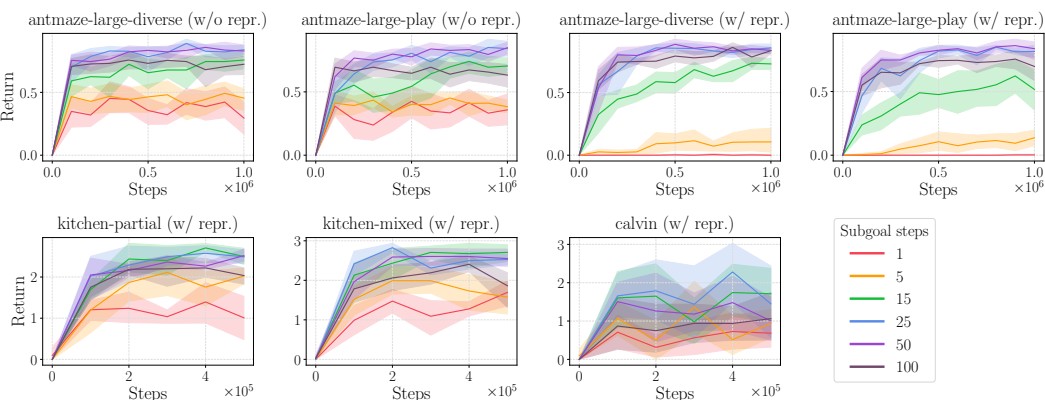

Figure 16: Ablation study of the subgoal steps $k$. HIQL generally achieves the best performances when $k$ is between 25 and 50. Even when $k$ is not within this range, HIQL mostly maintains reasonably good performance unless $k$ is too small (*i.e.*, $\leq 5$). Shaded regions denote the 95% confidence intervals across 8 random seeds.

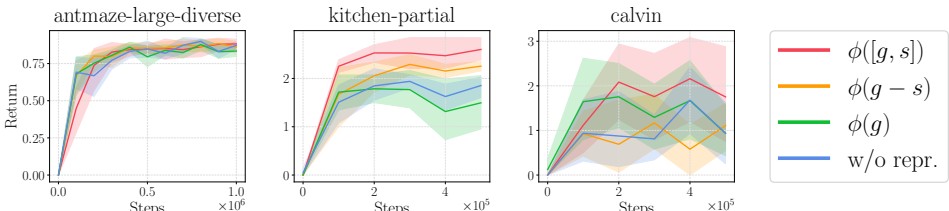

Figure 17: Ablation study of different parameterizations of the representation function. Passing $s$ and $g$ together to $\phi$ improves performance in general. Shaded regions denote the 95% confidence intervals across 8 random seeds.

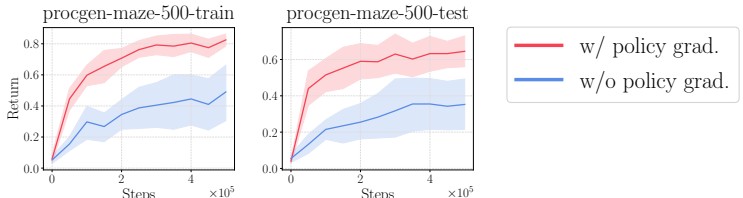

Figure 18: Ablation study of the auxiliary gradient flow from the low-level policy loss to $\phi$ on pixel-based ProcGen Maze. This auxiliary gradient flow helps maintain goal information in the representations. Shaded regions denote the 95% confidence intervals across 8 random seeds.

## C    Ablation Study

**Subgoal steps.**    To understand how the subgoal steps $k$ affect performance, we evaluate HIQL with six different $k \in \{1, 5, 15, 25, 50, 100\}$ on AntMaze, Kitchen, and CALVIN. On AntMaze, we test both HIQL with and without representations (Section 5.2). Figure 16 shows the results, suggesting that HIQL generally achieves the best performance with $k$ between 25 and 50. Also, HIQL still maintains reasonable performance even when $k$ is not within this optimal range, unless $k$ is too small.

**Representation parameterizations.**    We evaluate four different choices of the representation function $\phi$ in HIQL: $\phi([g, s])$, $\phi(g - s)$, $\phi(g)$, and without $\phi$. Figure 17 shows the results, indicating that passing $g$ and $s$ together to $\phi$ generally improves performance. We hypothesize that this is because $\phi$, when given both $g$ and $s$, can capture contextualized information about the goals (or subgoals) with respect to the current state, which is often easier to deal with for the low-level policy. For example, in AntMaze, the agent only needs to know the relative position of the subgoal with respect to the current position.

**Auxiliary gradient flows for representations.**    We found that in pixel-based environments (*e.g.*, Procgen Maze), allowing gradient flows from the low-level policy loss to the representation function improves performance (Figure 18). We believe this is because the additional gradients from the policy loss further help maintain the information necessary for control. We also (informally) found that this additional gradient flow occasionally slightly improves performances in the other environments as well, but we do not enable this feature in state-based environments to keep our method as simple as possible.

## D    Implementation details

We implement HIQL based on JaxRL Minimal [32]. Our implementation is available at the following repository: https://github.com/seohongpark/HIQL. We run our experiments on an internal GPU cluster composed of TITAN RTX and A5000 GPUs. Each experiment on state-based environments takes no more than 8 hours and each experiment on pixel-based environments takes no more than 16 hours.

## D.1 Environments

**AntMaze [9, 87]**  We use the 'antmaze-medium-diverse-v2', 'antmaze-medium-play-v2', 'antmaze-large-diverse-v2', and 'antmaze-large-play-v2' datasets from the D4RL benchmark [28]. For AntMaze-Ultra, we use the 'antmaze-ultra-diverse-v0' and 'antmaze-ultra-play-v0' datasets proposed by Jiang et al. [43]. The maze in the AntMaze-Ultra task is twice the size of the largest maze in the original D4RL dataset. Each dataset consists of 999 length-1000 trajectories, in which the Ant agent navigates from an arbitrary start location to another goal location, which does not necessarily correspond to the target evaluation goal. At test time, to specify a goal $g$ for the policy, we set the first two state dimensions (which correspond to the $x$-$y$ coordinates) to the target goal given by the environment and the remaining proprioceptive state dimensions to those of the first observation in the dataset. At evaluation, the agent gets a reward of 1 when it reaches the goal.

**Kitchen [35].**  We use the 'kitchen-partial-v0' and 'kitchen-mixed-v0' datasets from the D4RL benchmark [28]. Each dataset consists of 136950 transitions with varying trajectory lengths (approximately 227 steps per trajectory on average). In the 'kitchen-partial-v0' task, the goal is to achieve the four subtasks of opening the microwave, moving the kettle, turning on the light switch, and sliding the cabinet door. The dataset contains a small number of successful trajectories that achieve the four subtasks. In the 'kitchen-mixed-v0' task, the goal is to achieve the four subtasks of opening the microwave, moving the kettle, turning on the light switch, and turning on the bottom left burner. The dataset does not contain any successful demonstrations, only providing trajectories that achieve some subset of the four subtasks. At test time, to specify a goal $g$ for the policy, we set the proprioceptive state dimensions to those of the first observation in the dataset and the other dimensions to the target kitchen configuration given by the environment. At evaluation, the agent gets a reward of 1 whenever it achieves a subtask.

**CALVIN [63].**  We use the offline dataset provided by Shi et al. [84], which is based on the teleoperated demonstrations from Mees et al. [63]. The task is to achieve the four subtasks of opening the drawer, turning on the lightbulb, sliding the door to the left, and turning on the LED. The dataset consists of 1204 length-499 trajectories. In each trajectory, the agent achieves some of the 34 subtasks in an arbitrary order, which makes the dataset highly task-agnostic [84]. At test time, to specify a goal $g$ for the policy, we set the proprioceptive state dimensions to those of the first observation in the dataset and the other dimensions to the target configuration. At evaluation, the agent gets a reward of 1 whenever it achieves a subtask.

**Procgen Maze [16].**  We collect an offline dataset of goal-reaching behavior on the Procgen Maze suite. For each maze level, we pre-compute the optimal goal-reaching policy using an oracle, and collect a trajectory of 1000 transitions by commanding a goal, using the goal-reaching policy to reach this goal, then commanding a new goal and repeating henceforth. The 'procgen-maze-500' dataset consists of 500000 transitions collected over the first 500 levels and 'procgen-maze-1000' consists of 1000000 transitions over the first 1000 levels. At test time, we evaluate the agent on "challenging" levels that contain at least 20 leaf goal states (*i.e.*, states that have only one adjacent state in the maze). We use 50 such levels and goals for each evaluation, where they are randomly sampled either between Level 0 and Level 499 for the "-train" tasks or between Level 5000 and Level 5499 for the "-test" tasks. The agent gets a reward of 1 when it reaches the goal.

**Visual AntMaze.**  We convert the original state-based AntMaze environment into a pixel-based environment by providing both a $64 \times 64 \times 3$-dimensional camera image (as shown in the bottom row of Figure 6b) and 27-dimensional proprioceptive states without global coordinates. For the datasets, we use the converted versions of the 'antmaze-large-diverse-v2' and 'antmaze-large-play-v2' datasets from the D4RL benchmark [28] as well as a newly collected dataset, 'antmaze-large-navigate-v2', which consists of diverse navigation behaviors that visit multiple goal locations within an episode. The task and the evaluation scheme are the same as the original state-based AntMaze environment.

**Roboverse [25, 104].**  We use the same dataset and tasks used in Zheng et al. [104]. The dataset consists of 3750 length-300 trajectories,[1] out of which we use the first 3334 trajectories for training

---

[1]While Zheng et al. [104] separate each length-300 trajectory into four length-75 trajectories, we found that using the original length-300 trajectories improves performance in general.

(which correspond to approximately 1000000 transitions), while the remaining trajectories are used as a validation set. Each trajectory in the dataset features four random primitive behaviors, such as pushing an object or opening a drawer, starting from randomized initial object poses. At test time, we employ the same five goal-reaching tasks used in Zheng et al. [104]. We provide a precomputed goal image, and the agent gets a reward of 1 upon successfully completing the task by achieving the desired object poses.

In Tables 1 to 3, we report the normalized scores with a multiplier of 100 (AntMaze, Procgen Maze, Visual AntMaze, and Roboverse) or 25 (Kitchen and CALVIN).

## D.2 Hyperparameters

We present the hyperparameters used in our experiments in Table 4, where we mostly follow the network architectures and hyperparameters used by Ghosh et al. [34]. We use layer normalization [5] for all MLP layers. For pixel-based environments, we use the Impala CNN architecture [21] to handle image inputs, mostly with 512-dimensional output features, but we use normalized 10-dimensional output features for the goal encoder of HIQL's value function to make them easily predictable by the high-level policy, as discussed in Appendix A. We do not share encoders between states and goals, or between different components. As a result, in pixel-based environments, we use a total of *five* separate CNN encoders (two for the value function, two for the high-level policy, and two for the low-level policy, but the goal encoder for the value function is the same as the goal encoder for the low-level policy (Figure 1a)). In Visual AntMaze and Roboverse, we apply a random crop [48] (with probability 0.5) to prevent overfitting, following Zheng et al. [104].

During training, we periodically evaluate the performance of the learned policy at every 100K (state-based) or 50K (pixel-based) steps, using 52 (AntMaze, Kitchen, CALVIN, and Visual AntMaze), 50 (Procgen Maze), or 110 (Roboverse, 22 per each task) rollouts[2]. At evaluation, we use $\arg\max$ actions for environments with continuous action spaces and $\epsilon$-greedy actions with $\epsilon = 0.05$ for environments with discrete action spaces (*i.e.*, Procgen Maze). Following Zheng et al. [104], in Roboverse, we add Gaussian noise with a standard deviation of 0.15 to the $\arg\max$ actions.

To ensure fair comparisons, we use the same architecture for both HIQL and four baselines (GCBC, HGCBC, GC-IQL, and GC-POR). The discount factor $\gamma$ is chosen from $\{0.99, 0.995\}$, the AWR temperature $\beta$ from $\{1, 3, 10\}$, the IQL expectile $\tau$ from $\{0.7, 0.9\}$ for each method.

For HIQL, we set $(\gamma, \beta, \tau) = (0.99, 1, 0.7)$ across all environments. For GC-IQL and GC-POR, we use $(\gamma, \beta, \tau) = (0.99, 3, 0.9)$ (AntMaze-Medium, AntMaze-Large, and Visual AntMaze), $(\gamma, \beta, \tau) = (0.995, 1, 0.7)$ (AntMaze-Ultra), or $(\gamma, \beta, \tau) = (0.99, 1, 0.7)$ (others). For the subgoal steps $k$ in HIQL, we use $k = 50$ (AntMaze-Ultra), $k = 3$ (Procgen Maze and Roboverse), or $k = 25$ (others). HGCBC uses the same subgoal steps as HIQL for each environment, with the exception of AntMaze-Ultra, where we find it performs slightly better with $k = 25$. For HIQL, GC-IQL, and GC-POR, in state-based environments and Roboverse, we sample goals for high-level or flat policies from either the future states in the same trajectory (with probability 0.7) or the random states in the dataset (with probability 0.3). We sample high-level goals only from the future states in the other environments (Procgen Maze and Visual AntMaze).

# E Proofs

## E.1 Proof of Proposition 4.1

For simplicity, we assume that $T/k$ is an integer and $k \leq T$.

---

[2]These numbers include two additional rollouts for video logging (except for Procgen Maze).

Table 4: Hyperparameters.

| Hyperparameter | Value |
|---|---|
| # gradient steps | 1000000 (AntMaze), 500000 (others) |
| Batch size | 1024 (state-based), 256 (pixel-based) |
| Policy MLP dimensions | $(256, 256)$ |
| Value MLP dimensions | $(512, 512, 512)$ |
| Representation MLP dimensions (state-based) | $(512, 512, 512)$ |
| Representation architecture (pixel-based) | Impala CNN [21] |
| Nonlinearity | GELU [37] |
| Optimizer | Adam [47] |
| Learning rate | 0.0003 |
| Target network smoothing coefficient | 0.005 |

*Proof.* Defining $z_1 := z_{1,T}$ and $z_2 := z_{-1,T}$, the probability of the flat policy $\pi$ selecting an incorrect action can be computed as follows:

$$\mathcal{E}(\pi) = \mathbb{P}[\hat{V}(s+1, g) \leq \hat{V}(s-1, g)] \tag{12}$$

$$= \mathbb{P}[\hat{V}(1, T) \leq \hat{V}(-1, T)] \tag{13}$$

$$= \mathbb{P}[-(T-1)(1 + \sigma z_1) \leq -(T+1)(1 + \sigma z_2)] \tag{14}$$

$$= \mathbb{P}[z_1 \sigma(T-1) - z_2 \sigma(T+1) \leq -2] \tag{15}$$

$$= \mathbb{P}[z \sigma \sqrt{T^2 + 1} \leq -\sqrt{2}] \tag{16}$$

$$= \Phi\left(-\frac{\sqrt{2}}{\sigma \sqrt{T^2 + 1}}\right), \tag{17}$$

where $z$ is a standard Gaussian random variable, and we use the fact that the sum of two independent Gaussian random variables with standard deviations of $\sigma_1$ and $\sigma_2$ follows a normal distribution with a standard deviation of $\sqrt{\sigma_1^2 + \sigma_2^2}$.

Similarly, the probability of the hierarchical policy $\pi^\ell \circ \pi^h$ selecting an incorrect action is bounded using a union bound as

$$\mathcal{E}(\pi^\ell \circ \pi^h) \leq \mathcal{E}(\pi^h) + \mathcal{E}(\pi^\ell) \tag{18}$$

$$= \mathbb{P}[\hat{V}(s+k, g) \leq \hat{V}(s-k, g)] + \mathbb{P}[\hat{V}(s+1, s+k) \leq \hat{V}(s-1, s+k)] \tag{19}$$

$$= \mathbb{P}[\hat{V}(k, T) \leq \hat{V}(-k, T)] + \mathbb{P}[\hat{V}(1, k) \leq \hat{V}(-1, k)] \tag{20}$$

$$= \Phi\left(-\frac{\sqrt{2}}{\sigma \sqrt{(T/k)^2 + 1}}\right) + \Phi\left(-\frac{\sqrt{2}}{\sigma \sqrt{k^2 + 1}}\right). \tag{21}$$

□

## E.2 Proof of Proposition 5.1

We first formally define some notations. For $s \in \mathcal{S}, a \in \mathcal{A}, g \in \mathcal{S}$, and a representation function $\phi : \mathcal{S} \to \mathcal{Z}$, we denote the goal-conditioned state-value function as $V(s, g)$, the action-value function as $Q(s, a, g)$, the parameterized state-value function as $V_\phi(s, z)$ with $z = \phi(g)$, and the parameterized action-value function as $Q_\phi(s, a, z)$. We assume that the environment dynamics are deterministic, and denote the deterministic transition kernel as $p(s, a) = s'$. Accordingly, we have $Q(s, a, g) = V(p(s, a), g) = V(s', g)$ and $Q_\phi(s, a, z) = V_\phi(p(s, a), z) = V_\phi(s', z)$. We denote the optimal value functions with the superscript "*", *e.g.*, $V^*(s, g)$. We assume that there exists a parameterized value function, which we denote $V_\phi^*(s, \phi(g))$, that is the same as the true optimal value function, *i.e.*, $V^*(s, g) = V_\phi^*(s, \phi(g))$ for all $s \in \mathcal{S}$ and $g \in \mathcal{S}$.

*Proof.* For $\pi^*$, we have

$$\pi^*(a \mid s, g) = \underset{a \in \mathcal{A}}{\arg\max} \, Q^*(s, a, g) \tag{22}$$

$$= \underset{s' \in \mathcal{N}_s}{\arg\max} \, V^*(s', g) \tag{23}$$

$$= \underset{s' \in \mathcal{N}_s}{\arg\max} \, V_\phi^*(s', z), \tag{24}$$

where $\mathcal{N}_s$ denotes the neighborhood sets of $s$, *i.e.*, $\mathcal{N}_s = \{s' \mid \exists a, p(s, a) = s'\}$. For $\pi_\phi^*$, we have

$$\pi_\phi^*(a \mid s, z) = \underset{a \in \mathcal{A}}{\arg\max} \, Q_\phi^*(s, a, z) \tag{25}$$

$$= \underset{s' \in \mathcal{N}_s}{\arg\max} \, V_\phi^*(s', z). \tag{26}$$

By comparing Equation (24) and Equation (26), we can see that they have the same $\arg\max$ action sets for all $s$ and $g$. $\qquad\square$

# F  Subgoal Visualizations

We visualize learned subgoals in Figures 19 and 20 (videos are available at `https://seohong.me/projects/hiql/`). For AntMaze-Large, we train HIQL without representations and plot the $x$-$y$ coordinates of subgoals. For Procgen Maze, we train HIQL with 10-dimensional representations and find the maze positions that have the closest representations (with respect to the Euclidean distance) to the subgoals produced by the high-level policy. The results show that HIQL learns appropriate $k$-step subgoals that lead to the target goal.

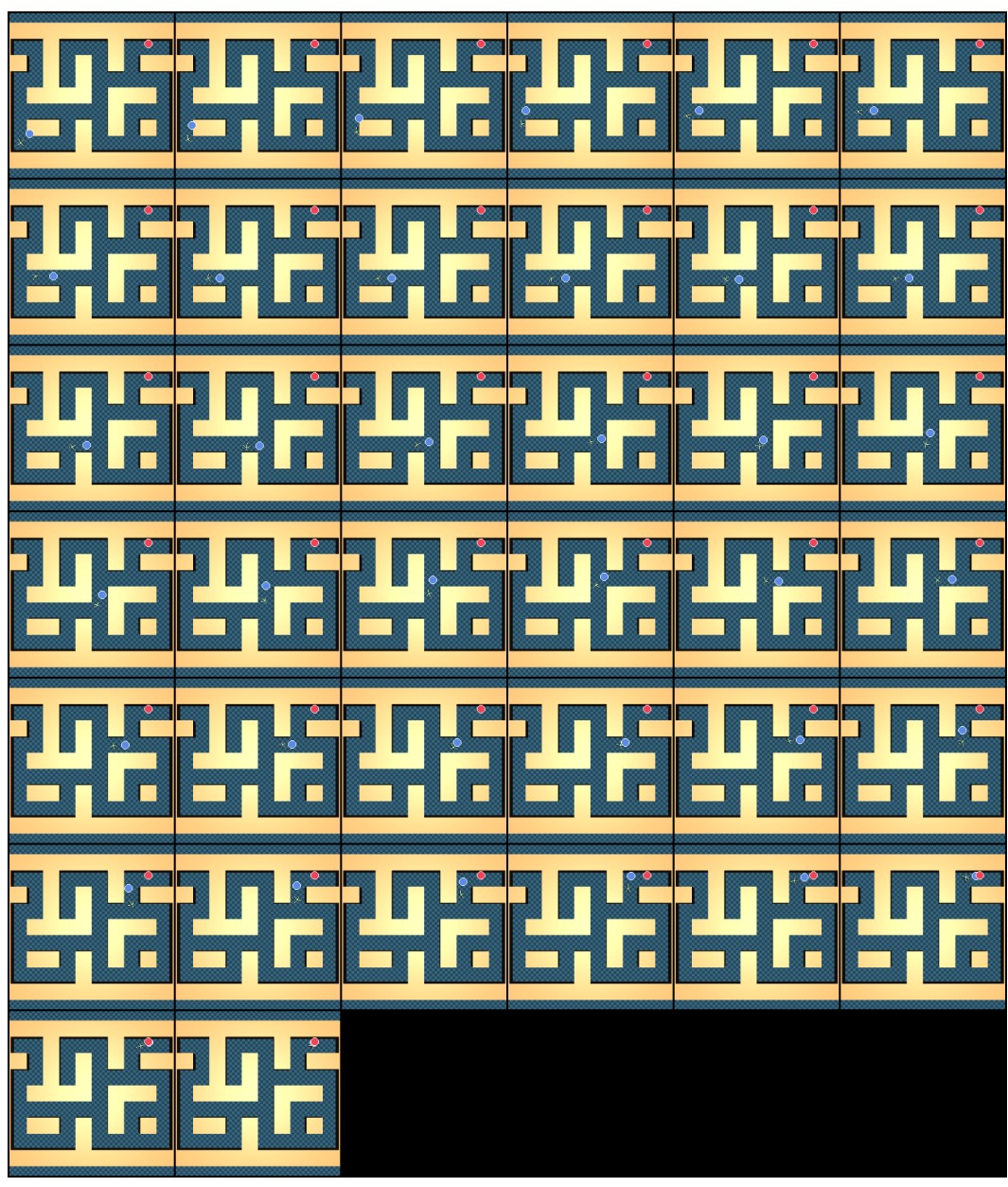

Figure 19: Subgoal visualization in AntMaze-Large. The red circles denote the target goal and the blue circles denote the learned subgoals. Videos are available at https://seohong.me/projects/hiql/.

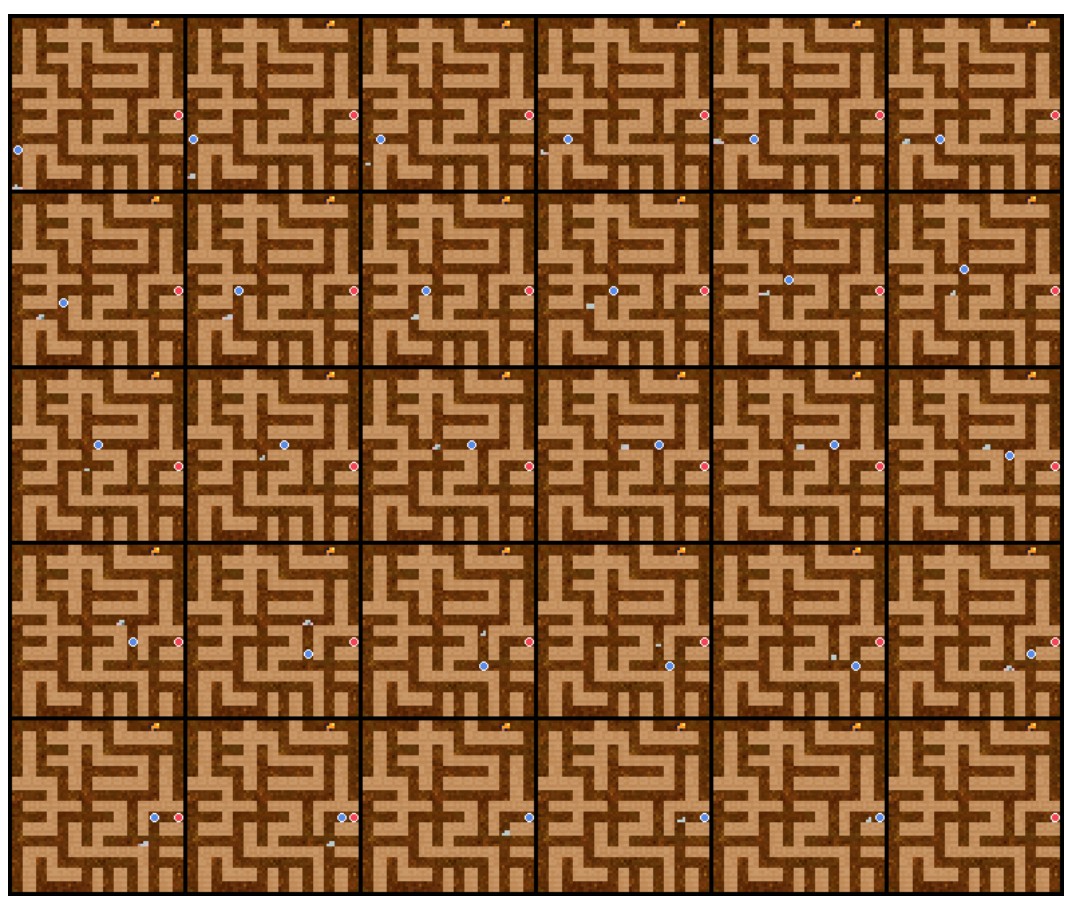

Figure 20: Subgoal visualization in Procgen Maze. The red circles denote the target goal, the blue circles denote the learned subgoals, and the white blobs denote the agent. Videos are available at https://seohong.me/projects/hiql/.

