# OpenReview forum: "HIQL: Offline Goal-Conditioned RL with Latent States as Actions"
_NeurIPS.cc/2023/Conference — NeurIPS 2023 spotlight_

### Official Review · Reviewer_kqhw · 2023-06-27

**Soundness:** 4 excellent
**Presentation:** 4 excellent
**Contribution:** 3 good
**Rating:** 7
**Confidence:** 4

**Summary:**

The paper introduces HIQL, a hierarchical algorithm for offline goal-conditioned RL. HIQL utilizes an action-free version of IQL to learn the value function and subsequently derives both high-level and low-level policies from this shared value function using AWR. The paper asserts that the hierarchical structure offers increased robustness to value function noise, leading to enhanced performance in achieving long-horizon goals. The effectiveness of the proposed method is validated through experiments conducted in both state-based and pixel-based tasks.

**Strengths:**

* Overall, the paper is well written and well organized.
* The proposed method HIQL incorporates IQL into a hierarchical framework, similar to prior offline RL work POR. However, HIQL incorporates additional designs from hierarchical RL, including predicting latent waypoints and k-step waypoints. Consequently, it is a new approach from both directions, offering valuable insights for tackling offline GCRL with hierarchical RL.
* The examples in Section 4.1 and Section 4.3 are helpful for readers to understand.
* The experiments are thorough, showing improved performance on both state-based and image-based tasks.


**Weaknesses:**

* The given examples show that the hierarchical structure is more robust to value noises in discrete tasks. Generally, in continuous tasks, the value function tends to be smoother. This raises the question of whether the performance improvement observed in HIQL is solely attributed to the hierarchical structure. If the claim in the paper is right, a method using smoothed value function (e.g., [1]) without hierarchical structure may achieve comparable performance with HIQL. Additional explanation and comparison are encouraged.
* More information to reproduce the Figure 2 need to be provided.
* In the appendix, the authors concat [s,g] before feeding into $\phi$. Therefore, it is recommended that Figure 1 be revised to address this matter.
* In the experiments, all the baselines considered are related to (weighted) imitation learning, making the inclusion of additional baselines utilizing temporal difference learning [2][3] necessary for a more comprehensive evaluation. In addition, it is notable that HIQL leverages the advantage of goal relabeling, while baselines such as IQL do not utilize this technique. Despite this, IQL has already demonstrated strong performance on the pixel-based Procgen Maze benchmark. Therefore, it remains unknown if HIQL can achieve comparable performance when compared to IQL+HER.
* For the experiments, I encourage the authors to report the performance of the low-level policy without the high-level policy. This can effectively highlight the true usefulness of the hierarchical structure.
* There is an error in Eq (7) in which the variable g is mistakenly used to train the low-level policy, while $s_{t+1}$ in the expectation is not used.

[1] Hong Z W, Yang G, Agrawal P. Bilinear value networks[J]. arXiv preprint arXiv:2204.13695, 2022.

[2] Chebotar Y, Hausman K, Lu Y, et al. Actionable models: Unsupervised offline reinforcement learning of robotic skills[J]. arXiv preprint arXiv:2104.07749, 2021.

[3] Li J, Tang C, Tomizuka M, et al. Hierarchical planning through goal-conditioned offline reinforcement learning[J]. IEEE Robotics and Automation Letters, 2022, 7(4): 10216-10223.

**Questions:**

* Additional explanation is needed to clarify how the hierarchical structure can be applicable in continuous situations, as well as a comparison with smoothed Q methods
* More information should be provided to enable the reproduction of Figure 2.
* Additional baselines for temporal difference learning and goal relabeling should be included in order to facilitate a more comprehensive and fair comparison.
* Authors are encouraged to report the performance of the low-level policy in order to emphasize the necessity of a hierarchical structure.
* Is it possible to expand the HIQL framework to include more than two levels? Considering that the parameter k is manually determined, the top level could be assigned a large k value, while a medium level would utilize a moderate k value.
* A fix in Figure 1 and Eq (7) may be needed.


## post rebuttal
I am delighted to see that the authors have addressed all of my concerns, and I am pleased to raise my score for this paper.

**Limitations:**

One limitation mentioned in the paper is the necessity for deterministic dynamics. It would be beneficial for the authors to further elaborate on why disentangling the controllable aspects from the uncontrollable elements in the environment is advantageous.

---

> ### Author Rebuttal · Authors · 2023-08-06
>
> We thank the reviewer for the thorough review and suggestions for improving the work. Below, we describe how we have revised the paper to clarify parts of the paper to address the questions raised by the reviewer. We also clarify how we have already compared to baselines that use HER, and present new results from three additional baselines. We believe that these changes strengthen the paper, and welcome additional suggestions for further improving the work.
>
> - **“It remains unknown if HIQL can achieve comparable performance when compared to IQL+HER”**
>
> We would like to clarify that “IQL” in this paper always refers to “**IQL+HER**” (i.e., goal-conditioned IQL) (L297) (and likewise “POR” in this paper always denotes “POR + HER”), and they use the exactly **same** goal relabeling strategy as HIQL. This relabeling strategy (described in Appendix A) is directly taken from a prior work without modification [15]; thus it is not tuned for our method, which ensures a fair comparison. To further clarify that the baselines are also goal conditioned and employ hindsight relabeling, we will refer to them as “GC-IQL” and “GC-POR” in the final version of the paper.
>
> * **Additional baselines that do not use (weighted) imitation learning**
>
> Thank you for the suggestion. We have now compared HIQL with **three** additional baselines that do not use (weighted) imitation learning.
>
> As we were unable to find publicly available official implementations of Actionable Models (AM) [17] or HiGOC [9], we tried to evaluate AM using both our own re-implementation and another re-implementation by Ma et al. [14]. However, despite our extensive efforts to tune its hyperparameters on two different codebases, we were unable to achieve performance above 1% even on the simplest AntMaze-Medium tasks. Hence, we instead evaluated the performance of **goal-conditioned CQL** (“GC-CQL”) [13], another temporal difference learning-based method that uses a similar [17] conservative objective to AM. For a fair comparison, we used the same hindsight relabeling strategy for GC-CQL and tuned its hyperparameters individually for each D4RL task.
>
> Moreover, we made additional comparisons with two recent offline goal-conditioned RL methods (Contrastive RL [11] and GCPC [2]) that report their performances on the goal-conditioned variants of D4RL tasks (AntMaze and/or Kitchen). **Contrastive RL** is a goal-conditioned value learning method based on contrastive learning, and **GCPC** is an offline goal-conditioned RL method that models trajectories using a BERT-style Transformer with masking techniques. For these methods, we directly used the reported results from the respective papers [2, 11].
>
> We present the full comparison results (averaged over 4 seeds) on D4RL environments in **Table 1** in the supplementary PDF. The results show that HIQL mostly achieves the best performance in these tasks, outperforming (or at least being as good as) the new baselines.
>
> * **Flat IQL with a smoothed value function**
>
> As the reviewer pointed out, a regularized (smoothed) value function may also alleviate the signal-to-noise challenge in a continuous state space. To empirically verify this hypothesis, we evaluated the performance of flat goal-conditioned IQL (GC-IQL) with a bilinear value function $V(s, g) = f(s)^\top \psi(g)$ with 512-dimensional $f$ and $\psi$.
>
> We present the results across four different tasks in **Table 2** in the supplementary PDF. Despite the improved smoothness of the value function, Bilinear GC-IQL shows worse performance than the original GC-IQL. We believe this is due to the limited expressivity of bilinear value functions compared to the original monolithic value functions. We will include a discussion about continuous state spaces in the final version of the paper.
>
> * **Performance of the low-level policy without a high-level policy**
>
> As per the reviewer’s suggestion, we evaluated the performance of HIQL’s low-level policy (without a high-level policy) on D4RL tasks. **Table 3** in the supplementary PDF presents the results across four different tasks. The results show that HIQL without a high-level policy almost completely fails to solve these tasks. This is because the low-level policy is trained to reach only nearby goals (Eq. (7)). If we train the low-level policy with the full set of goals, it becomes equivalent to GC-IQL, whose performance still falls behind that of HIQL. This highlights the necessity of our hierarchical structure.
>
> * **Is it possible to expand the HIQL framework to include more than two levels?**
>
> Thank you for raising this point. As the reviewer mentioned, it is indeed possible to have a recursive structure, in which higher-level policies produce subgoals for the policies at the next level down, and only the lowest-level policy produces actions. At an earlier stage of this research, we tested 3- or 4-level hierarchies on AntMaze-Large with waypoint steps of (25, 5) or (100, 25, 5). However, we found that they do not significantly improve (or sometimes hurt) performance compared to the two-level policy structure. This is likely because (1) a two-level policy is sufficient for AntMaze-Large given its episode length, and (2) policy errors can accumulate as the hierarchy grows. However, we believe such a recursive structure has the potential to solve much longer-horizon problems (where an improved signal-to-noise ratio outweighs accumulated policy errors), and further extending HIQL in this way with a highly scalable architecture is an exciting future research direction.
>
> * **Code to reproduce Figure 2**
>
> We have uploaded the ipynb file that we used to produce Figure 2 to our anonymized repository.
>
> * **Improving Figure 1, Typo in Equation (7)**
>
> Thank you for the suggestions! We have addressed the issue with Figure 1 and fixed the typo in Equation (7) in a revised version of the paper.
>
> Please let us know if there are any additional concerns or questions.
>
> **References**: Please see our global response.

---

> > ### Comment · Reviewer_kqhw · 2023-08-11
> > **Thank you for the response**
> >
> > Thank you for providing such a detailed response. After thoroughly reading your explanation, I am pleased to say that all of my concerns have been addressed. The clarification of "IQL+HER" and the additional comparisons have significantly strengthened the paper's persuasiveness. I found the discussion on the multi-level hierarchical structure particularly intriguing, and I hope to see it included in the revised version of the paper. Overall, I am now inclined to raise my score to 7, and I am delighted to see this work accepted by the conference.

---

### Official Review · Reviewer_ZY68 · 2023-06-28

**Soundness:** 3 good
**Presentation:** 4 excellent
**Contribution:** 3 good
**Rating:** 7
**Confidence:** 4

**Summary:**

This paper introduces a hierarchical approach to address the issue of offline goal-conditioned problems, specifically when certain trajectories in the dataset have missing actions. The proposed method tackles this challenge by simultaneously learning a value function with a modified version of IQL and extracting the high-level and low-level policies of the learned value function. The high-level policy predicts the representation of waypoints, while the low-level policy, conditioned on the waypoint, determines the primitive actions. Extensive experimental results on offline goal-conditioned tasks demonstrate the effectiveness of the proposed approach.


**Strengths:**

1. The writing is very easy to follow.
2. The paper proposes an elegant approach to goal-conditioned offline RL problem by decomposing the high-level and low-level policy while sharing the same value functions V.
3. The approach taken in this paper allows the adoption of the existing large amount of action-free data.
4. Good empirical performance on a comprehensive set of evaluation environments with both state and pixel observations.



**Weaknesses:**

For some of the design choices adopted in this paper, I think there are some alternatives in the literature that are good to compare with. See my comment in the question section.

**Questions:**

1. (**Clarification Question**) I am confused by the notation $V(s, \phi(g))$, $\pi(a|s, \phi(g))$. I understand that for the high-level policy, the waypoint it predicted is the latent representation, not the pixel values of the image. But is $s$ also represented by the same state representation/encoder $\phi$? So is it $V(\phi(s), \phi(g))$ (or with some stop gradient on either $\phi(s)$ or $\phi(g)) $?
2. (**How to learn the value function $V$**) In this paper, the author adapts the approach of the IQL algorithm to learn the value function V (with some modification to account for missing actions). But this is not the only approach to learn the value function. [1] proposes to learn the value-function V by dualizing the objective, which also does not require action labels in the dataset. Maybe it would be nice to compare this approach and see which approach gives better quality of the value function, since it is crucial for the extraction of high-quality policies.

[1] Ma et al. VIP: Towards Universal Visual Reward and Representation via Value-Implicit Pre-Training

---

> ### Author Rebuttal · Authors · 2023-08-06
>
> We thank the reviewer for the thorough review and constructive feedback about this work.
>
> * **How to encode $s$ in $V(s, \phi(g))$?**
>
> For pixel-based environments, we encode $s$ and $g$ separately into $\psi(s)$ and $\phi(g)$ using two *different* CNNs, $\psi$ and $\phi$. We then concatenate $\psi(s)$ to $\phi(g)$ and model $V(\psi(s), \phi(g))$ using an MLP. (For state-based environments, we directly concatenate $s$ to $\phi(g)$, without having a separate encoder for $s$.) Since only the $\phi(g)$ component is used outside as a goal representation, we simply denoted $V(\psi(s), \phi(g))$ as $V(s, \phi(g))$ in the paper (treating $\psi(s)$ as a “black box” within $V$). However, we recognize that this could have led to some confusion, and we will clarify this point in the final version of the paper.
>
> * **Alternatives to IQL for learning a goal-conditioned value function**
>
> As the reviewer pointed out, our hierarchical policy extraction scheme is *orthogonal* to the choice of the underlying offline RL algorithm used to learn a goal-conditioned value function $V(s, g)$. In this work, we chose to use IQL for its effectiveness and simplicity, as it does not require any other additional components and is easy to use. However, it is indeed possible to combine HIQL with other value-based offline RL algorithms, such as VIP [10], Contrastive RL [11], or Quasimetric RL [12] (or even CQL [13] or GoFAR [14] if we have action labels), and we believe our hierarchical policy structure can still be beneficial in these cases, as our theoretical results do not depend on the underlying value learning algorithm. We will clarify this point in the final paper. We believe studying different underlying offline RL algorithms that can further enhance this idea is an interesting direction for future research.
>
> Please let us know if there are any additional concerns or questions.
>
> **References**: Please see our global response.

---

> > ### Comment · Reviewer_ZY68 · 2023-08-10
> > **Thanks!**
> >
> > Thanks for the response. I appreciate the additional experiments on two pixel-based environments as well as including more baselines into the comparison. I will be sticking to the deserved high score that I have given the paper.

---

### Official Review · Reviewer_cj3e · 2023-07-03

**Soundness:** 4 excellent
**Presentation:** 3 good
**Contribution:** 3 good
**Rating:** 7
**Confidence:** 3

**Summary:**

The paper identifies an issue with offline goal-conditioned reinforcement learning: namely that goal-conditioned value estimation can be noisy, so long-horizon tasks  can be difficult to accomplish due to accumulating errors in value estimation. In addition, when states are so close together, there is very little signal for learning as any mistakes can be corrected in future states. Thus, HIQL is proposed, which learns a high-level goal-conditioned policy to predict subgoals (such that there is sufficient learning signal), and a low-level goal conditioned policy to predict actions, which benefits from only considering nearby goals. Experiments show the advantages of this approach with respect to baseline methods in hierarchical and flat offline RL, as well as the ability to use unlabeled (without action) trajectory data to improve learning, and data efficiency.

**Strengths:**

- The argument is extremely clear and well exposed
- I appreciate the demonstration of issues with current methods (Fig. 2, 3, 4) before introducing the solution, which helps my understanding of the field in general. Such a simple demonstration, though perhaps not exactly well validated, may even be worth more as expository writing than the final proposed method.
- The method is simple, in particular reuse of internal representations instead of learning some VAE or something more complex is nice.
- The ability to have different data requirements for different parts of the policy is quite nice, given that action labels are harder to come by (Table 3).

**Weaknesses:**

- Though the exposition is very clear, sometimes it is a bit repetitive. In particular, the same justification for hierarchy is pointed out in line 36 and 160, as well as the repetition between sections 4.2 and section 5, though I can see how this is a matter of personal taste.
- The difference in domain between Figure 2 and Figure 3 is a little bit odd, though this isn't a huge issue. The domain in Figure 3 is maybe a bit _too_ toy, which leads me to discount Proposition 4.1, the major point was already in Figure 2.
- I'm a bit suspect of the claim that Procgen Maze is a good demonstration for a visual environment, given that the visual observation is already so structured. It would be nice to work on a more complex task within Procgen, like Coinrun, though I'm unaware of the data availability for these tasks.
- Proposition 5.1 is unnecessary in my opinion, that or just a simple proof sketch could be given inline.
- I think a more in-depth discussion of particularly related work is merited. Novelty is not very clear to me from the given presentation. In particular the proposed method is so simple that it's surprising to me that it does not already exist.

**Questions:**

Besides the points in the weakness section, it's a bit unclear to me the novelty of this submission, so I would appreciate some further contextualization to other hierarchical methods that take advantage of data.

**Limitations:**

Limitations are addressed.

---

> ### Author Rebuttal · Authors · 2023-08-06
>
> We thank the reviewer for the thorough review and constructive feedback about this work.
>
> * **“Novelty is not very clear to me from the given presentation.”**
>
> One major difference between our approach and prior hierarchical methods is that we extract both policies (and even representations) from a **single** (non-hierarchical) value function. This eliminates the need for additional components, unlike previous hierarchical methods that train separate value functions [4, 5] or use potentially complex high-level subgoal planning procedures [6, 7, 8, 9]. Despite its simplicity, perhaps surprisingly, we show that this simple technique can significantly improve performance both in theory and in practice, due to an improved “signal-to-noise” ratio (Section 4). We have clarified this point in both the Introduction and Related Work sections in a revised version of the draft.
>
> * **Additional pixel-based environments**
>
> To verify the effectiveness of HIQL in more diverse visual environments beyond Procgen Maze, we evaluated HIQL and prior methods on **two** additional pixel-based benchmarks: Roboverse and Visual AntMaze. **Roboverse** [3] is a pixel-based, goal-conditioned robotic manipulation task that requires multi-stage reasoning and generalization, where the agent must learn to control a robot arm to manipulate objects purely from pixels. We use the same dataset and tasks used by Zheng et al. [3]. **Visual AntMaze** is a vision-based variant of the AntMaze environment, where we provide the agent with a $64 \times 64 \times 3$ camera image and its proprioceptive states, excluding the global coordinates. Hence, the agent must learn to navigate the maze based on the wall structure and floor color from the image. Please find illustrations of these environments in **Figure 1** of the supplementary PDF. For the datasets, we render the ‘antmaze-large-diverse-v2’ and ‘antmaze-large-play-v2’ datasets from the D4RL benchmark. We additionally employ a more challenging dataset, ‘antmaze-large-navigate-v2’, which consists of diverse navigation behaviors that visit multiple goal locations within an episode.
>
> In these two additional pixel-based environments, we compare the performances of HIQL (ours), goal-conditioned IQL (“GC-IQL”), goal-conditioned POR (“GC-POR”), HGCBC, and GCBC (with individually tuned hyperparameters). We report the results (averaged over 8 seeds, $\pm$ denotes standard deviations) below:
>
> | Task | GCBC | HGCBC (+ repr.) | GC-IQL | GC-POR (+ repr.) | **HIQL (ours)** |
> |---|---:|---:|---:|---:|---:|
> | visual-antmaze-diverse | $71.4 \pm 6.0$ | $35.1 \pm 12.0$ | $72.6 \pm 5.9$ | $47.4 \pm 17.6$ | $\mathbf{80.5} \pm 9.4$ |
> | visual-antmaze-play | $64.4 \pm 6.3$ | $23.8 \pm 8.5$ | $70.4 \pm 26.6$ | $57.0 \pm 8.1$ | $\mathbf{78.4} \pm 4.6$ |
> | visual-antmaze-navigate | $33.2 \pm 7.9$ | $21.4 \pm 4.6$ | $22.1 \pm 14.1$ | $16.1 \pm 15.2$ | $\mathbf{45.7} \pm 18.1$ |
> | roboverse | $26.2 \pm 4.5$ | $26.4 \pm 6.4$ | $31.2 \pm 8.7$ | $46.6 \pm 7.4$ | $\mathbf{61.5} \pm 5.3$ |
>
> The table above shows that HIQL outperforms previous methods in these vision-based tasks as well. Notably, in Roboverse, HIQL is capable of generalizing to solve unseen robotic manipulation tasks purely from images, achieving an average success rate of 62%. We have included these results and experimental details in a revised version of the paper.
>
> * **Writing suggestions**
>
> Thank you for the suggestions! We will incorporate them into the final version of the paper.
>
> Please let us know if there are any additional concerns or questions.
>
> **References**: Please see our global response.

---

> > ### Comment · Reviewer_cj3e · 2023-08-10
> >
> > I thank the authors for their clarification on novelty, especially as I am somewhat unfamiliar with the prior work. They should take my original comment on surprise at the method being novel as a compliment. Simplicity is always appreciated.
> >
> > I also appreciate the additional visual domain experiments, which are much more convincing to me than the original maze domain. I see that the conclusions in the original paper remain undisturbed.
> >
> > I don't have any additional concerns at this time, and I think the original score of 7 that I gave is merited.

---

### Official Review · Reviewer_xrL8 · 2023-07-07

**Soundness:** 4 excellent
**Presentation:** 4 excellent
**Contribution:** 2 fair
**Rating:** 7
**Confidence:** 5

**Summary:**

This paper introduces hierarchical IQL (HIQL) for offline goal-conditioned reinforcement learning. HIQL uses the IQL algorithm to learn both a high-level waypoint policy as well as a low-level action policy; in both case, the goal-conditioned value is provided by the same value function learned using goal-conditioned IQL procedure. HIQL is evaluated on both state-based as well as image-based environments that require hierarchical planning. In all settings, HIQL outperforms prior methods

**Strengths:**

Originality:
This paper builds on IQL, which has already been extended to goal-conditioned settings, though not in the hierarchical manner this paper proposes. Furthermore, the use of single value function is a nice idea. The didactic examples and theoretical results are also interesting and contribute to the overall story. Therefore, I believe this paper contains enough original components for NeurIPS.

Quality and Clarity:
This paper is very well written and presented. The experiments are rigorously executed and explained.

Significance:
Offline GCRL is an emerging and important topic to study, and this paper proposes a conceptually simple and effective algorithm for the setting.

**Weaknesses:**

This is a strong paper without major weaknesses. There are several places to further improve the paper:

1. An ablation that demonstrates that AWR is needed for both policies. I suspect that it could be possible that it is not super important for the low-level controller if the horizon is short enough.

2. Connection with existing approaches that learns a high-level planner and uses an inverse dynamics model to back out low-level actions can be better drawn in related work.

3. I believe more substantive error bound in the form of policy performance can be derived in addition to Proposition 4.1; [1] makes one such attempt.

[1] Ajay, Anurag, et al. "Opal: Offline primitive discovery for accelerating offline reinforcement learning." arXiv preprint arXiv:2010.13611 (2020).



**Questions:**

I have listed several suggestions in the section above. Overall, I think this is a solid paper, and I recommend acceptance.

**Limitations:**

N/A.

---

> ### Author Rebuttal · Authors · 2023-08-06
>
> We thank the reviewer for the thorough review and constructive feedback about this work. As suggested by the reviewer, we ran an additional ablation experiment to study the use of AWR for both policies.
>
> * **An ablation that demonstrates that AWR is needed for both policies**
>
> We ablated both low-level AWR and high-level AWR by replacing either of them with BC, and evaluated them on both AntMaze, which has a relatively large waypoint step ($k=25$ for ‘-large’ and $k=50$ for ‘-ultra’), and Procgen Maze, which has a relatively small waypoint step ($k=3$). We report the performance below (the results are averaged over 4 seeds and $\pm$ denotes standard deviations).
>
> | Task                   | HIQL (ours) (High AWR, Low AWR) | Ablation (High AWR, Low BC) | Ablation (High BC, Low AWR) |
> |------------------------|--------------------------|-----------------------------|-----------------------------|
> | antmaze-large-diverse  | $88.2 \pm 5.3$           | $60.1 \pm 13.3$             | $\mathbf{91.3} \pm 5.1$     |
> | antmaze-ultra-diverse  | $\mathbf{52.9} \pm 17.4$ | $23.6 \pm 9.2$              | $47.6 \pm 13.7$             |
> | procgen-maze-500-train | $\mathbf{82.5} \pm 6.0$  | $74.0 \pm 12.6$             | $17.5 \pm 5.7$              |
> | procgen-maze-500-test  | $\mathbf{64.5} \pm 13.2$ | $52.0 \pm 11.7$             | $19.5 \pm 9.6$              |
>
> The table above shows that both high-level AWR and low-level AWR are important for performance. However, their relative importance may depend on the dataset. In AntMaze, low-level AWR is more important than high-level AWR (due to the data collection strategy; the AntMaze datasets consist of single-goal-reaching trajectories with noisy actions, in which case BC is a reasonable objective for the high-level policy but not for the low-level policy). On the contrary, in Procgen Maze, high-level AWR is more important than low-level AWR (due to both the diversity of the dataset and the relatively small waypoint steps).
>
> * **Connection with existing approaches that learn a high-level planner and use an inverse dynamics model to back out low-level actions**
>
> Thank you for the suggestion. We will discuss these approaches [18, 19, 20, 21, 22] in the related work section of the final version.
>
> * **An additional error bound based on suboptimality**
>
> Thank you for the suggestion. In an earlier version of the draft, we made initial attempts at deriving a bound similar to that of Ajay et al., but found removing the assumption of two value functions (we only use one) to be difficult. If we are able to successfully derive a hierarchical performance bound based on a single value function, we will include it in the final paper.
>
> Please let us know if there are any additional concerns or questions.
>
> **References**: Please see our global response.

---

> > ### Comment · Reviewer_xrL8 · 2023-08-15
> > **Thank You for Your Response**
> >
> > Dear Authors,
> >
> > Thank you for your response. I have carefully read it and am satisfied with the new experimental results that justify the design decision of using AWR at both levels. Given the initial high rating, I will keep my original score as I believe that this paper merits acceptance at NeurIPS.

---

### Official Review · Reviewer_tvKa · 2023-07-08

**Soundness:** 3 good
**Presentation:** 3 good
**Contribution:** 3 good
**Rating:** 7
**Confidence:** 4

**Summary:**

The paper proposes HIQL a hierarchical algorithm for offline goal-conditioned RL. The approach consists in utilizing a single action-free value function to acquire knowledge about the structure and employ two policies: a high-level policy that predicts or represents a waypoint, and a low-level policy that predicts the action required to reach that waypoint. The approach is well explained and motivated. The experiments comparing most state of the art approaches of offline-RL on the main benchmarks seem convincing.

**Strengths:**

Reasonably novel approach, well explained and illustrated with convincing results on main offline RL benchmark of locomotion and manipulation against reasonable baselines.

**Weaknesses:**

Maybe performing TT on kitchen as baseline would have been useful to improve the comparison part.
Otherwhile, no major weakness in my opinion.

**Questions:**

Why not using the TT in the kitchen scenario for comparison ?

**Limitations:**

Nothing noticeable IMO.

---

> ### Author Rebuttal · Authors · 2023-08-06
>
> We thank the reviewer for the thorough review and constructive feedback about this work.
>
> * **Why not use TT in Kitchen for comparison?**
>
> As mentioned in L303, we directly took the performances of the Trajectory Transformer (TT) and Trajectory Autoencoding Planner (TAP) from Jiang et al. [1], where these methods were not evaluated on the Kitchen benchmark. As such, we only made a comparison with these methods on AntMaze-{Medium, Large, Ultra}, which Jiang et al. [1] also used as a benchmark. We chose to use the numbers from prior work directly, since setting up and tuning these Transformer-based methods often requires a substantial amount of time and extensive computing resources.
>
> However, to address this comment, we additionally compare HIQL with GCPC [2], a recently proposed offline goal-conditioned RL method that also uses a Transformer to model trajectories similarly to TT. GCPC shows a better performance than TT in AntMaze-Large and reports its performance on goal-conditioned Kitchen (but not on AntMaze-Ultra). We present the results below, where we took the GCPC results from the original paper [2]:
>
>
> | Task                   | GCBC            | GC-IQL          | TAP    | TT               | GCPC [2]        | HIQL (ours)              |
> |------------------------|-----------------|-----------------|--------|------------------|-----------------|--------------------------|
> | antmaze-medium-diverse | $67.3 \pm 10.1$ | $63.5 \pm 14.6$ | $85.0$ | $\mathbf{100.0}$ | $70.8$          | $86.8 \pm 4.6$           |
> | antmaze-medium-play    | $71.9 \pm 16.2$ | $70.9 \pm 11.2$ | $78.0$ | $\mathbf{93.3}$  | $70.4$          | $84.1 \pm 10.8$          |
> | antmaze-large-diverse  | $20.2 \pm 9.1$  | $50.7 \pm 18.8$ | $82.0$ | $60.0$           | $77.2$          | $\mathbf{88.2} \pm 5.3$  |
> | antmaze-large-play     | $23.1 \pm 15.6$ | $56.5 \pm 14.4$ | $74.0$ | $66.7$           | $79.2$          | $\mathbf{86.1} \pm 7.5$  |
> | antmaze-ultra-diverse  | $14.4 \pm 9.7$  | $21.6 \pm 15.2$ | $26.0$ | $33.3$           | -               | $\mathbf{52.9} \pm 17.4$ |
> | antmaze-ultra-play     | $20.7 \pm 9.7$  | $29.8 \pm 12.4$ | $22.0$ | $20.0$           | -               | $\mathbf{39.2} \pm 14.8$ |
> | kitchen-partial        | $38.5 \pm 11.8$ | $39.2 \pm 13.5$ | -      | -                | $\mathbf{65.0}$ | $\mathbf{65.0} \pm 9.2$  |
> | kitchen-mixed          | $46.7 \pm 20.1$ | $51.3 \pm 12.8$ | -      | -                | $61.0$          | $\mathbf{67.7} \pm 6.8$  |
>
> The table above shows that, despite its simplicity, HIQL achieves the best (including ties) performance in both antmaze-large-{diverse, play} and kitchen-{partial, mixed}, generally outperforming the more computationally intensive Transformer-based methods. Putting it all together, we hope that the omission of TT/TAP results on Kitchen (which was not evaluated with these methods in prior work) is reasonable.
>
> Please let us know if there are any additional concerns or questions.
>
> **References**: Please see our global response.

---

### Author Rebuttal · Authors · 2023-08-06

We appreciate all five reviewers’ constructive feedback and suggestions for improving the work. We would like to highlight the updates we made in our responses below.
- We evaluated HIQL and baselines on $\mathbf{2}$ **additional pixel-based environments**, Roboverse and Visual AntMaze, which demonstrate the effectiveness of HIQL in more diverse visual domains (Reviewer cj3e).
- We compared HIQL with $\mathbf{3}$ **additional baselines** — goal-conditioned CQL, contrastive RL, and GCPC — on D4RL tasks (Reviewers tvKa and kqhw).
- We ablated various aspects of HIQL, which shows the necessity of both high-level AWR and low-level AWR and the necessity of having a high-level policy (Reviewers xrL8 and kqhw).

Below are the references we use in our responses:

[1] Jiang et al., Efficient Planning in a Compact Latent Action Space. ICLR 2023.

[2] Zeng et al., Goal-Conditioned Predictive Coding as an Implicit Planner for Offline Reinforcement Learning. arXiv 2023.

[3] Zheng et al., Stabilizing Contrastive RL: Techniques for Offline Goal Reaching. arXiv 2023.

[4] Nachum et al., Data-Efficient Hierarchical Reinforcement Learning. NeurIPS 2018.

[5] Levy et al., Learning Multi-Level Hierarchies with Hindsight. ICLR 2019.

[6] Shah et al., Rapid Exploration for Open-World Navigation with Latent Goal Models. CoRL 2021.

[7] Fang et al., Planning to Practice: Efficient Online Fine-Tuning by Composing Goals in Latent Space. IROS 2022.

[8] Fang et al., Generalization with Lossy Affordances: Leveraging Broad Offline Data for Learning Visuomotor Tasks. CoRL 2022.

[9] Li et al., Hierarchical Planning Through Goal-Conditioned Offline Reinforcement Learning. RA-L 2022.

[10] Ma et al., VIP: Towards Universal Visual Reward and Representation via Value-Implicit Pre-Training. ICLR 2023.

[11] Eysenbach et al., Contrastive Learning as Goal-Conditioned Reinforcement Learning. NeurIPS 2022.

[12] Wang et al., Optimal Goal-Reaching Reinforcement Learning via Quasimetric Learning. ICML 2023.

[13] Kumar et al., Conservative Q-Learning for Offline Reinforcement Learning. NeurIPS 2020.

[14] Ma et al., How Far I'll Go: Offline Goal-Conditioned Reinforcement Learning via f-Advantage Regression. NeurIPS 2022.

[15] Ghosh et al., Reinforcement Learning from Passive Data via Latent Intentions. ICML 2023.

[16] Hong et al., Bilinear value networks. ICLR 2022.

[17] Chebotar et al., Actionable Models: Unsupervised Offline Reinforcement Learning of Robotic Skills. ICML 2021.

[18] Torabi et al., Behavioral Cloning from Observation. IJCAI 2018.

[19] Schmeckpeper et al., Reinforcement Learning with Videos: Combining Offline Observations with Interaction. CoRL 2020.

[20] Baker et al., Video PreTraining (VPT): Learning to Act by Watching Unlabeled Online Videos. NeurIPS 2022.

[21] Chang et al., Learning Value Functions from Undirected State-only Experience. ICLR 2022.

[22] Zheng et al., Semi-Supervised Offline Reinforcement Learning with Action-Free Trajectories. ICML 2023.

---

### Decision · Program_Chairs · 2023-09-21

**Decision:**

Accept (spotlight)

**Comment:**

The paper proposes goal-conditioned off-line RL as general (pre-) training strategy and introduces a hiearchical variant. The reviewers appreciated novelty, theoretical results, descriptions of the shortcomings of the SOTA, clarity and presentation, execution and experiments.

Only minor weaknesses had been identified, mostly linked to some additional ablations and additional connections to related work. Overally, a strong and well-rounded paper has been identified.

The AC concurs and recommends acceptance.